# SmoothHess: ReLU Network Feature Interactions via Stein's Lemma

**Max Torop,**[1][*] **Aria Masoomi,**[1][*] **Davin Hill,**[1] **Kivanc Kose,**[2] **Stratis Ioannidis,**[1] **Jennifer Dy**[1]

[1] Northeastern University, [2] Memorial Sloan Kettering Cancer Center

{torop.m, masoomi.a}@northeastern.edu,
{dhill, ioannidis, jdy}@ece.neu.edu,
{kosek}@mskcc.org

## Abstract

Several recent methods for interpretability model feature interactions by looking at the Hessian of a neural network. This poses a challenge for ReLU networks, which are piecewise-linear and thus have a zero Hessian almost everywhere. We propose *SmoothHess*, a method of estimating second-order interactions through Stein's Lemma. In particular, we estimate the Hessian of the network convolved with a Gaussian through an efficient sampling algorithm, requiring only network gradient calls. SmoothHess is applied post-hoc, requires no modifications to the ReLU network architecture, and the extent of smoothing can be controlled explicitly. We provide a non-asymptotic bound on the sample complexity of our estimation procedure. We validate the superior ability of SmoothHess to capture interactions on benchmark datasets and a real-world medical spirometry dataset.

## 1 Introduction

As machine learning models are increasingly relied upon in a variety of high-stakes applications such as credit lending [50, 42] medicine [10, 67, 15], or law [38], it is important that users are able to interpret model predictions. To this end, many methods have been developed to assess the importance of individual input features in effecting model output [53, 76, 81, 66, 73]. However, one may achieve a deeper understanding of model behavior by quantifying how features *interact* to affect predictions. While diverse notions of feature interaction have been proposed [53, 57, 56, 63, 25, 77, 84], in this work, we focus on the intuitive partial-derivative definition of feature interaction [25, 4, 47, 85, 18, 39].

Given a function and point, the Interaction Effect [47] between a given set of features on the output is the partial derivative of the function output with respect to the features; intuitively, it represents the infinitesimal change in the function engendered by a joint infinitesimal change in each chosen feature. The Interaction Effect derives in part from Friedman and Popescu [25], who define the *global* interaction between a set of features over the data distribution as the expected square of the partial-derivative with respect to those features. As in prior works [47, 39], we eschew the expectation to focus on *local* interactions occurring around a given point $x$ and avoid squaring partial-derivatives to maintain the directionality of the interaction. We focus on *pair-wise feature interactions*, which, in the context of the Interaction Effect, are the elements of the Hessian.

ReLU networks are a popular class of neural networks that use ReLU activation functions [28]. The use of ReLU has desirable properties, such as the mitigation of the vanishing gradient problem [28], and it is the sole activation function used in popular neural network families such as ResNet [32] and VGG [72]. However, ReLU networks are piece-wise linear [58] and thus have a zero Hessian almost everywhere (a.e.), posing a problem for quantifying interactions (see Figure 1(a)).

---

[*]Equal contribution

37th Conference on Neural Information Processing Systems (NeurIPS 2023).

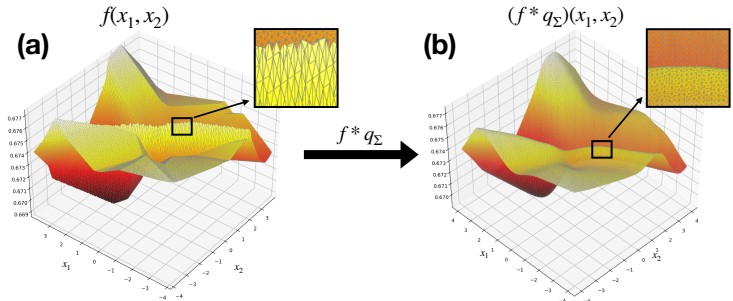

Figure 1: **(a)** An exemplar illustration of a simple 5-hidden-layer ReLU network $f : \mathbb{R}^2 \to \mathbb{R}$. Note that $f$ is piece-wise linear and thus has $\nabla_x^2 f(x) = 0$ a.e. **(b)** The ReLU network convolved with $q_{0.3I} : \mathbb{R}^2 \to \mathbb{R}$, the density function of $\mathcal{N}(0, 0.3I)$. This function is no longer piece-wise linear and admits non-zero higher order derivatives.

A common approach for estimating the Hessian in ReLU networks is to take the Hessian of a smooth surrogate network which approximates the original: each ReLU is replaced with SoftPlus, a smooth approximation to ReLU [28, 22], before differentiating [39, 20]. However, this approach affords one only coarse control over the smoothing as *each internal neuron* is smoothed, leading to unwieldy asymmetric effects, as can be seen in Figure 2.

In this work, we propose *SmoothHess*: the Hessian of the ReLU network convolved with a Gaussian. Such a function is a more flexible smooth surrogate than SoftPlus as *smoothing is done on the network output*, and the covariance of the Gaussian allows one to mediate the contributions of points based on their Mahalanobis distance. Unfortunately, obtaining the Hessian of the convolved ReLU network is impossible using naïve Monte-Carlo (MC) averaging. However, by proving an extension of Stein's Lemma [78, 49], we show that such a quantity can be efficiently estimated using *only gradient calls* on the original network. For an illustration of ReLU network smoothing, see Figure 1(b).

Our **main contributions** are as follows:

- We propose *SmoothHess*, the Hessian of the Gaussian smoothing of a neural network, as a model of the second-order feature interactions.
- We derive a variant of Stein's Lemma, which allows one to estimate SmoothHess for ReLU networks using only gradient calls.
- We prove non-asymptotic sample complexity bounds for our SmoothHess estimator.
- We empirically validate the superior flexibility of SmoothHess to capture interactions on MNIST, FMNIST, and CIFAR10. We utilize SmoothHess to derive insights into a network trained on a real-world medical spirometry dataset. Our code is publicly available.[*]

The remainder of this paper is organized as follows: In Sec. 2, we summarize the gradient-based methods for feature importance and interactions that are most related to our work. In Sec. 3, we provide a technical preliminary covering the definitions and techniques underlying both our method and competing works. Next, in Sec. 4, we introduce our method, SmoothHess, explain how to estimate it, and provide sample complexity bounds for our estimation procedure. In Sec. 5, we experimentally demonstrate the ability of SmoothHess to model interactions. Finally, in Sec. 6, we summarize our method and results, followed by a discussion of limitations and possible solutions.

## 2 Related Work

**Feature Importance and First-Order Methods:** Methods that quantify feature importance fall into two categories: (i) perturbation-based methods (e.g., [53, 66, 17]), which evaluate the change in model outputs with respect to perturbed inputs, and (ii) gradient-based methods (e.g., [72, 76, 81]), which leverage the natural interpretation of the gradient as infinitesimally local importance for a given sample. Most relevant to our work are gradient-based approaches. The saliency map, as defined in [72], is simply the gradient of model output with respect to the input. Several variants are developed to address the shortcomings of the saliency maps. SmoothGrad [76] was developed

---

[*]https://github.com/MaxTorop/SmoothHess

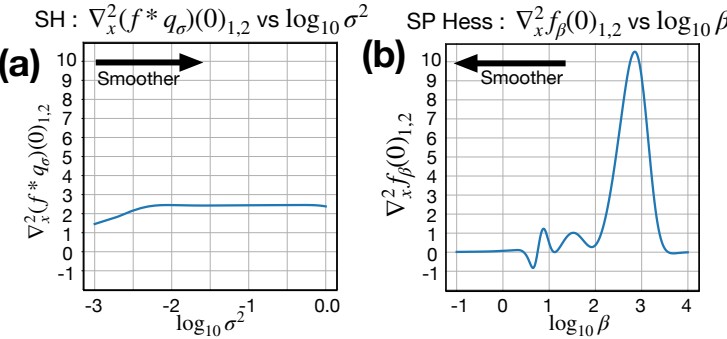

Figure 2: Estimated Hessian element between features 1 and 2 at $x_0 = (0,0)^T$ for a 6-layer ReLU Network $f : \mathbb{R}^2 \to \mathbb{R}$ trained to memorize the Four Quadrant toy dataset. **(a)** SmoothHess (SH) is estimated with isotropic covariance $\Sigma = \sigma^2 I$ using granularly sampled $\sigma^2 \in \{1e\text{-}3, \ldots, 1\}$. Aside from at minute $\sigma^2 < 1e\text{-}2.5$, where hyper-local noisy behavior is captured, we have $\nabla^2(f * q_\sigma)(x_0)_{1,2} \approx (5 + 3 + 12 - 10)/4 = 2.5$, the average of the memorized "ground truth" off-diagonal Hessian element over the four quadrants. This indicates a symmetry in the weighting of the contributions from points around $x_0$ at *every level of smoothing*. **(b)** The Hessian of the SoftPlus smoothed function $f_\beta$ (SP Hess) is computed using granularly sampled $\beta \in \{1e\text{-}1, \ldots, 1e4\}$. The average value of 2.5 is not achieved at any value of $\beta$, aside from briefly between $\log_{10} \beta = 2$ and $\log_{10} \beta = 3.5$ indicating that SoftPlus fails to incorporate the information around $x_0$ in a symmetric manner at *every level of smoothing*.

to address noise by averaging saliency maps (see also Sec. 3), and comes with sample complexity guarantees [3]. Sundararajan et al. [81] introduce Integrated Gradients, the path integral between an input and an uninformative baseline. This is extended to the Shapley framework by Erion et al. [24]. The Grad-CAM line of work [93, 70, 61] is similar in nature to the methods above, with the key distinction that importance is modeled over internal (hidden) layers.

**Feature Interactions:** A variety of methods have been proposed to estimate higher-order feature interactions, which can again be separated into perturbation-based [54, 82, 86, 57, 92] and gradient-based approaches. Among the latter, Tsang et al. [85] propose Gradient-NID, which estimates feature interaction strength as the corresponding Hessian element squared. Janizek et al. [39] propose Integrated Hessian, which extends Integrated Gradients to use a path-integrated Hessian. Lerman et al. [47] propose Taylor-CAM, a higher-order generalization of Grad-Cam [70]. Cui et al. [18] quantify global interactions for Bayesian neural networks in terms of the expected value of the Hessian over the data distribution. For classification ReLU networks, the CASO and CAFO explanation vectors exploit feature interactions in the loss function using Hessian estimates [74].

Unfortunately, due to their piecewise linearity, existing Hessian-based interaction methods cannot be readily applied to ReLU networks. Janizek et al. [39] replace each ReLU activation with SoftPlus post-hoc, before applying Integrated Hessians. Similarly, Tsang et al. [85] apply their method to networks with the SoftPlus activation instead of ReLU. For regression tasks, Lerman et al. [47] replace ReLU with the smooth activation function GELU [34] before training. Although SoftMax outputs and the cross-entropy loss admit higher order derivatives, pre- or post-hoc smoothing, as above, is necessary for finding interactions affecting logits, internal neurons, or regression outputs. Indeed, while Singla et al. [74] estimate interactions on the original ReLU network, they are only with respect to the loss function.

In contrast, we propose a method for quantifying feature interactions that works with any ReLU network post-hoc without requiring retraining or modifications to the network architecture. It also can be directly estimated with respect to model *as well as* intermediate layer outputs. Furthermore, our experiments in Sec. 5 show the superior ability of our method to model logit outputs, internal neurons, and SoftMax probabilities as compared to a SoftPlus smoothed network.

## 3 Technical Preliminary

**ReLU Network Background:** We denote a ReLU network by $F : \mathbb{R}^d \to \mathbb{R}^c$, where $c = 1$ in the case of regression. We denote the function which we wish to explain as $f : \mathbb{R}^d \to \mathbb{R}$, an arbitrary neuron $f_i^{(l)}$ (the $i^{th}$ neuron in the $l^{th}$ layer) in our ReLU network, or a SoftMax Probability

for some class $k \in \{1, \ldots, c\}$. We denote sample space $\mathcal{X} \subseteq \mathbb{R}^d$ and point $x_0 \in \mathcal{X}$ for which we wish to capture feature interactions affecting $f(x_0)$. A general $L$-hidden-layer ReLU Network $F = f^{(L+1)} : \mathbb{R}^d \to \mathbb{R}^c$ may be defined recursively by [33, 58]:

$$f^{(l)}(x_0) = W^{(l)} g^{(l-1)}(x_0) + b^{(l)}, \text{ for } l = 1 \ldots L + 1, \tag{1a}$$

$$g^{(l)}(x_0) = \max(0, f^{(l)}(x_0)), \text{ for } l = 1 \ldots L, \tag{1b}$$

with $g^{(0)}(x_0) = x_0 \in \mathbb{R}^d$. For a given layer $l$ the dimension (number of neurons) is defined as $n_l \in \mathbb{N}$, i.e. $f^{(l)}(x_0), g^{(l)}(x_0) \in \mathbb{R}^{n_l}$ (with $n_0 = d$) with weight and bias $W^{(l)} \in \mathbb{R}^{n_l, n_{l-1}}$, $b^{(l)} \in \mathbb{R}^{n_l}$. As stated above, ReLU networks are piecewise linear [58]. Specifically, *each neuron* $f_i^{(l)} : \mathbb{R}^d \to \mathbb{R}$, $l \in \{1, \ldots, L+1\}, i \in \{1, \ldots, n_l\}$, is a piecewise linear function, corresponding to a finite set of $K \in \mathbb{N}$ convex polytopes $\mathcal{Q} = \{Q_i\}_{i=1}^K$, $Q_i \subseteq \mathbb{R}^d$ which form a partition of $\mathbb{R}^d$ [30, 31, 58, 33, 71, 64, 7]. See Figure 1(a) for an example of a ReLU network.

**SmoothGrad:** SmoothGrad [76] is an extension of the saliency map $\nabla_x f(x_0)$ [72] developed to reduce noise. SmoothGrad is an average over saliency maps; formally, it is $\mathbb{E}_\delta[\nabla_x f(x_0 + \delta)]$ where $\delta \sim \mathcal{N}(0, \Sigma)$ and the covariance $\Sigma \in \mathbb{R}^{d \times d}$ is a hyperparameter usually chosen to be isotropic $\Sigma = \sigma^2 I, \sigma > 0$. SmoothGrad is estimated by sampling $n$ perturbation vectors $\{\delta_i\}_{i=1}^n$, $\delta_i \sim \mathcal{N}(0, \Sigma), \forall i \in \{1, \ldots, n\}$ and averaging via $\hat{G}_n^{\text{SG}}(x_0, f, \Sigma) = \frac{1}{n} \sum_{i=1}^n \nabla_x f(x_0 + \delta_i)$. Unfortunately, the analogous Hessian average $\frac{1}{n} \sum_{i=1}^n \nabla_x^2 f(x_0 + \delta_i) \approx \mathbb{E}_\delta[\nabla_x^2 f(x_0 + \delta)]$ is not useful for quantifying feature interactions in ReLU networks as $\mathbb{E}_\delta[\nabla_x^2 f(x_0 + \delta)] = 0$.

**SoftPlus:** A common approach to assessing higher-order derivatives for ReLU networks is by differentiating a smooth surrogate $f_\beta : \mathbb{R}^d \to \mathbb{R}$ where a SoftPlus replaces every ReLU in $f$ [28, 39, 20, 21]. The SoftPlus function $s_\beta(x_0) = \frac{1}{\beta} \log(1 + e^{\beta x_0})$, $\beta > 0$, is a smooth approximation of ReLU, where $\beta$ is a parameter inversely proportional to the level of smoothing [28, 22]. SoftPlus approaches ReLU and $f_\beta$ and $\nabla f_\beta$ approximate $f$ and $\nabla f$ with arbitrary accuracy as $\beta \to \infty$ [20]. Dombrowski et al. [20] establish empirical and theoretical connections between $\nabla_x f_\beta(x_0)$ and SmoothGrad. They observe visual similarities between $\nabla_x f_\beta(x_0)$ and SmoothGrad at appropriate $\beta$ values. Further, in the simple case of a single neuron with no bias, they prove that replacing ReLU with SoftPlus corresponds to a Gaussian-like smoothing of the gradient. However, this theory does not extend to networks with multiple neurons and layers.

**Stein's Lemma:** Stein's Lemma [78] is central to our analysis. We relate a variant of Stein's Lemma [51], which extends the results of Stein [78] to hold for multivariate normal distributions with arbitrary covariance matrices:

**Lemma 1.** *(Stein's Lemma [51]) Given* $x_0 \in \mathbb{R}^d$, *covariance matrix* $\Sigma \in \mathbb{R}^{d \times d}$, *multivariate normal random vector* $\delta \in \mathbb{R}^d$ *distributed from* $\delta \sim \mathcal{N}(0, \Sigma)$ *and almost everywhere differentiable function* $g : \mathbb{R}^d \to \mathbb{R}$ *for which* $\mathbb{E}_\delta[|[\nabla_x g(x_0 + \delta)]_i|] < \infty$ *for each* $i \in \{1, \ldots, d\}$, *then*

$$\mathbb{E}_\delta[\Sigma^{-1} \delta g(x_0 + \delta)] = \mathbb{E}_\delta[\nabla_x g(x_0 + \delta)]. \tag{2}$$

A zero-th order oracle associated with a function $g : \mathbb{R}^d \to \mathbb{R}$ is an oracle which, when provided with any input $x \in \mathbb{R}^d$, returns $g(x)$. Likewise, a first order oracle (or gradient oracle) returns $\nabla_x g(x)$. In the context of this work, where $g$ is a ReLU network, zero-th order and first order oracle calls amount to network forward passes and backpropagation calls, respectively.

Given one has access to $g$ as a zero-th order oracle, the LHS of Stein's Lemma may be estimated by sampling a set of perturbations $\{\delta_i\}_{i=1}^n; \delta_i \sim \mathcal{N}(0, \Sigma)$, querying the zero-th order oracle $g(x_0 + \delta_i)$ for each $\delta_i$, and MC-estimating, i.e:

$$\hat{G}_n(x_0, g, \Sigma) = \frac{1}{n} \sum_{i=1}^n \Sigma^{-1} \delta_i g(x_0 + \delta_i) \approx \mathbb{E}_\delta[\Sigma^{-1} \delta g(x_0 + \delta)] \overset{\text{Lemma 1}}{=} \mathbb{E}_\delta[\nabla_x g(x_0 + \delta)]. \tag{3}$$

Such an approach is useful for estimating $\mathbb{E}_\delta[\nabla_x g(x_0 + \delta)]$ in the RHS of Eq. (3) when the gradient of $g$ is impossible or expensive to obtain but $g$ itself may be efficiently queried as a zero-th order oracle (see e.g., [44, 48, 60]). Extending Liu [51]'s work, Lin et al. [49] present a second-order variant of Stein's Lemma expressing the expected Hessian in terms of the gradient:

**Lemma 2.** *(First-Order Oracle Stein's Lemma [49]) Given $x_0 \in \mathbb{R}^d$, covariance matrix $\Sigma \in \mathbb{R}^{d \times d}$, multivariate normal random vector $\delta \in \mathbb{R}^d$ distributed from $\delta \sim \mathcal{N}(0, \Sigma)$ and continuously differentiable function $g(z) : \mathbb{R}^d \to \mathbb{R}$ with locally Lipschitz\* gradients $\nabla g : \mathbb{R}^d \to \mathbb{R}^d$, then*

$$\mathbb{E}_\delta[\Sigma^{-1}\delta[\nabla_x g(x_0 + \delta)]^T] = \mathbb{E}_\delta[\nabla_x^2 g(x_0 + \delta)]. \tag{4}$$

Complexity bounds have been derived for similar identities, which express the Hessian using zero-th order information [6, 94, 23]. However, Lemma 2 *fails* for ReLU networks. This is precisely because ReLU networks are piecewise linear and, therefore, *are not continuously differentiable*. In the next section, we directly address this through our method for estimating a smoothed Hessian for ReLU networks.

# 4 SmoothHess

Our main contributions are: (1) We propose *SmoothHess*, the Hessian of the network convolved with a Gaussian, for modeling feature interactions. (2) We use Stein's Lemma to prove that SmoothHess may be estimated for ReLU networks using only gradient oracle calls. (3) We prove non-asymptotic sample complexity bounds for our SmoothHess estimator.

**Gaussian Convolution as a Smooth Surrogate:** An alternative smooth-surrogate to $f_\beta$ is $h_{f,\Sigma} : \mathbb{R}^d \to \mathbb{R}$, the convolution of $f$ with a Gaussian:

$$h_{f,\Sigma}(x_0) = (f * q_\Sigma)(x_0) = \int_{z \in \mathbb{R}^d} f(z) q_\Sigma(z - x_0) dz, \tag{5}$$

where $\Sigma \in \mathbb{R}^{d \times d}$ is a covariance matrix, $q_\Sigma(z - x_0) = (2\pi)^{-\frac{d}{2}} |\Sigma|^{-\frac{1}{2}} \exp(-\frac{1}{2}d(x_0, z)_\Sigma)$ is the density function of the Gaussian distribution $\mathcal{N}(0, \Sigma)$, $|\cdot|$ is the determinant and $d(x_0, z)_\Sigma = (x_0 - z)^T \Sigma^{-1} (x_0 - z) \in \mathbb{R}$ is the Mahalanobis distance between $x_0$ and $z$.

The Gaussian-smoothed function $h_{f,\Sigma}$ is infinitely differentiable and does not suffer from the limitations of surrogates obtained from internal smoothing. Here, smoothing is done on the *output image* of $f$ and thus the relationship between $h_{f,\Sigma}(x_0)$, $f(z)$, $z$ and $x_0$ is made explicit by Eq. (5): the relative contribution of $f(z)$ to $h_{f,\Sigma}(x_0)$ is proportional to the exponentiated negative half Mahalanobis distance $-\frac{1}{2}d(x_0, z)_\Sigma$. The ability to adjust $\Sigma$ gives a user fine-grained and localized control over smoothing; the eigenvectors and eigenvalues of $\Sigma$ respectively encode directions of input space and a corresponding locality for their contribution to $h_{f,\Sigma}$.

We define SmoothHess as the Hessian of $h_{f,\Sigma}$:

**Definition 1.** *(SmoothHess) Given ReLU network $f : \mathbb{R}^d \to \mathbb{R}$, point to explain $x_0 \in \mathbb{R}^d$, covariance matrix $\Sigma \in \mathbb{R}^{d \times d}$ and $q_\Sigma : \mathbb{R}^d \to \mathbb{R}$, the density function of Gaussian distribution $\mathcal{N}(0, \Sigma)$, then SmoothHess is defined to be the Hessian of $f$ convolved with $q_\Sigma$ evaluated at $x_0$:*

$$\nabla_x^2 h_{f,\Sigma}(x_0) = \nabla_x^2 (f * q_\Sigma)(x_0) = \int_{z \in \mathbb{R}^d} f(z) \nabla_x^2 q_\Sigma(z - x_0) dz. \tag{6}$$

Well-known properties of the Gaussian distribution may be used to encode desiderata into the convolved function and, accordingly, to SmoothHess through the choice of the covariance. For instance, it is known that as $d \to \infty$ the isotropic Gaussian distribution $\mathcal{N}(0, \sigma^2 I_{d \times d})$ converges to $U(S_{\sigma\sqrt{d}}^{d-1})$, a uniform distribution over the radius $\sigma\sqrt{d}$ sphere [88]. Thus, given large enough $d$, as is commonly encountered in deep learning datasets, one may choose $\Sigma = (r/\sqrt{d})I$ to approximately ensure that $h_{f,\Sigma}(x_0)$ incorporates information from the radius $r$ sphere around $x_0$. We exploit and validate this intuition in our experiments (see Table 1).

Finally, we must highlight the strong connection between SmoothHess and SmoothGrad [76], which Wang et al. [89] prove is equivalent to the gradient of the same smooth surrogate: $\nabla_x h_{f,\Sigma}(x_0)$. Thus,

---

\*In its most general form, Lemma 2 holds for functions which are continuously differentiable and have *locally ACL* gradients. Locally ACL functions are functions which are absolutely continuous on almost every straight line, a mild condition which is satisfied by both locally Lipschitz and continuously differentiable functions [49, 46, 68]. As ReLU networks are locally Lipschitz [29], they are locally ACL.

SmoothGrad and SmoothHess together define a second-order Taylor expansion of $h_{f,\Sigma}$ at $x_0$, which can be used as a second-order model of $f$ around $x_0$.

**SmoothHess Computation via Stein's Lemma:** We relate our method for estimating SmoothHess for ReLU networks. As stated above, Lemma 2 *does not hold for ReLU networks.* However, we extend the arguments of Wang et al. [89] and Lin et al. [49] to show that the LHS from Lemma 2 *is equivalent to* SmoothHess for all Lipschitz continuous functions[*]:

**Proposition 1.** *Given $x_0 \in \mathbb{R}^d$, L-Lipschitz continuous function $g : \mathbb{R}^d \to \mathbb{R}$, covariance matrix $\Sigma \in \mathbb{R}^{d \times d}$ and random vector $\delta \in \mathbb{R}^d$ distributed from $\delta \sim \mathcal{N}(0, \Sigma)$ with density function $q_\Sigma : \mathbb{R}^d \to \mathbb{R}$, then*

$$\mathbb{E}_\delta[\Sigma^{-1}\delta[\nabla_x g(x_0 + \delta)]^T] = \nabla_x^2[(g * q_\Sigma)(x_0)] = \nabla_x^2 h_{g,\Sigma}(x_0), \tag{7}$$

*where $*$ denotes convolution.*

In other words, even though Lemma 2 does not hold for ReLU networks, Stein's Lemma is indeed evaluating a Hessian: namely, SmoothHess given by Eq. (6). The proof consists of moving the Hessian operator on the RHS of Eq. (7) into the integral defined by $(g * q_\Sigma)(x_0)$. The resulting expression is simplified into a form for which Lin et al. [49] prove is equivalent to the LHS of Eq. (7). The proof is provided in App. B.

Proposition 1 opens up the possibility of an MC-estimate of SmoothHess *that only require access to a first-order oracle $\nabla f$*. This is computed by sampling a set of $n \in \mathbb{N}$ perturbations $\{\delta_i\}_{i=1}^n, \delta_i \sim \mathcal{N}(0, \Sigma)$, and for each $\delta_i$ querying $\nabla_x f(x_0 + \delta_i)$ before taking the outer product $\delta_i[\nabla_x f(x_0 + \delta_i)]^T \in \mathbb{R}^{d \times d}$, Monte-Carlo averaging and finally symmetrizing:

$$\hat{H}_n^\circ(x_0, f, \Sigma) = \frac{1}{n}\sum_{i=1}^n \Sigma^{-1}\delta_i[\nabla_x f(x_0 + \delta_i)]^T, \tag{8a}$$

$$\hat{H}_n(x_0, f, \Sigma) = \frac{1}{2}(\hat{H}_n^\circ(x_0, f, \Sigma) + \hat{H}_n^{\circ T}(x_0, f, \Sigma)). \tag{8b}$$

Each first-order oracle call has the same time complexity as one forward pass and may be computed efficiently in batches using standard deep learning frameworks [62, 1, 13]. As $n$ is finite, $\hat{H}_n^\circ(x_0, f, \Sigma)$ is not guaranteed to be symmetric in practice. Thus, we symmetrize our estimator in Eq. (8b). A straightforward consequence of Proposition 1 is that $\lim_{n \to \infty} \hat{H}_n(x_0, f, \Sigma) = \nabla_x^2 h_{f,\Sigma}(x_0)$, which we formally show in the Proof of Theorem 1 in App. C.

Estimation of SmoothGrad may be amortized with SmoothHess, obtaining $\nabla_x h_{f,\Sigma}(x_0)$ *at a significantly reduced cost*. The main computational expense when estimating SmoothGrad is the querying of the first-order oracle. As this querying is part of SmoothHess estimation, the gradients which we compute may be averaged at a minimal additional cost of $\mathcal{O}(nd)$ to obtain a SmoothGrad estimate. Likewise, SmoothHess may be obtained at a reduced cost of $\mathcal{O}(nd^2)$, the cost of the outer products, during SmoothGrad estimation. For details of our algorithm, see App. D.

Last, we prove non-asymptotic bounds for our SmoothHess estimator:

**Theorem 1.** *Let $f : \mathbb{R}^d \to \mathbb{R}$ be a piece-wise linear function over a finite partition of $\mathbb{R}^d$. Let $x_0 \in \mathbb{R}^d$, and denote $\{\delta_i\}_{i=1}^n$, a set of $n$ i.i.d random vectors in $\mathbb{R}^d$ distributed from $\delta_i \sim \mathcal{N}(0, \Sigma)$. Given $\hat{H}_n(x_0, f, \Sigma)$ as in Eq. (8), for any fixed $\varepsilon, \gamma \in (0, 1]$, given $n \geq \frac{4}{\varepsilon^2}[\max((C^+\sqrt{d} + \sqrt{\frac{1}{c^+}\log\frac{4}{\gamma}})^2, (C^-\sqrt{d} + \sqrt{\frac{1}{c^-}\log\frac{4}{\gamma}})^2)]$ then*

$$\mathbb{P}\left(\left\|\hat{H}_n - H\right\|_2 > \varepsilon\right) \leq \gamma, \tag{9}$$

*where $H = \nabla_x^2[(f * q_\Sigma)(x_0)]$, $C^+, C^- c^+, c^- > 0$ are constants depending on the function $f$ and covariance $\Sigma$ and $q_\Sigma : \mathbb{R}^d \to \mathbb{R}$ is the density function of $\mathcal{N}(0, \Sigma)$.*

We elect to use the non-asymptotic bounds presented in Theorem 3.39 of Vershynin [87], which hold for sums of outer products of a sub-gaussian random vector with itself. This poses a challenge as $\hat{H}_n$ is the sum of outer products of two sub-gaussian random vectors which are *not necessarily equal*. To deal with this issue, we separately prove non-asymptotic bounds for the outer products

---

[*]All ReLU network outputs, internal neurons, and SoftMax probabilities are Lipschitz continuous [29, 26].

| Dataset | MNIST | | | | FMNIST | | | | CIFAR10 | | | |
|---|---|---|---|---|---|---|---|---|---|---|---|---|
| *Function* | Class Logit (↓) | | Int. Neuron (↓) | | Class Logit (↓) | | Int. Neuron (↓) | | Class Logit (↓) | | Int. Neuron (↓) | |
| $\epsilon$ | 0.25 0.50 1.00 | | 0.25 0.50 1.00 | | 0.25 0.50 1.00 | | 0.25 0.50 1.00 | | 0.25 0.50 1.00 | | 0.25 0.50 1.00 | |
| SH+SG (Us) | **9.6e-7 7.8-6 6.7e-5** | | **4.9e-8 4.0e-7 3.3e-6** | | **6.5e-7 4.0e-6 4.3e-5** | | **2.0e-8 1.8e-7 1.6e-6** | | **9.8e-4 2.2e-2 1.2e-1** | | **8.1e-7 1.4e-5 1.6e-4** | |
| SG [76] | 4.5e-6 4.1e-5 3.9e-4 | | 2.1e-7 1.7e-6 1.5e-5 | | 3.0e-6 2.7e-5 2.6e-4 | | 1.0e-7 9.0e-7 7.0e-6 | | 1.3e-2 8.6e-2 4.9e-1 | | 1.3e-5 1.1e-4 8.3e-4 | |
| SP (H + G) | 1.2e-6 9.6e-6 8.1e-5 | | 5.5e-8 4.4e-7 3.7e-6 | | 9.6e-7 7.5e-6 6.5e-5 | | 3.0e-8 2.1e-7 1.8e-6 | | 2.1e-3 3.3e-2 2.5e-1 | | 1.1e-5 1.0e-4 7.0e-4 | |
| SP G | 4.6e-6 4.1e-5 3.9e-4 | | 2.1e-7 1.7e-6 1.5e-5 | | 3.2e-6 2.8e-5 2.6e-4 | | 1.0e-7 8.5e-7 7.2e-6 | | 1.3e-2 9.0e-2 5.2e-1 | | 5.1e-5 2.9e-4 1.6e-3 | |
| G [73] | 4.2e-3 1.7e-2 6.7e-2 | | 2.0e-3 7.0e-3 2.9e-2 | | 3.8e-3 1.5e-2 6.0e-2 | | 1.0e-4 4.0e-4 1.8e-3 | | 3.0e-1 1.2e-0 5.0e-0 | | 9.0e-4 3.5e-3 1.4e-2 | |

Table 1: Average $\mathcal{P}_{MSE}$ at three radii $\epsilon$. Results are calculated for SmoothHess + SmoothGrad (SH+SG,Us) SmoothGrad (SG) SoftPlus Gradient (SP G) SoftPlus Hessian + SoftPlus Gradient (SP (H + G)) and the vanilla Gradient (G). Results are provided for the predicted class logit, and the penultimate neuron maximally activated by the "three," dress and cat classes for MNIST, FMNIST and CIFAR10 respectively. While SP (H + G) and SP G results are reported using the best $\beta$ *chosen from over* 100 *values* based on validation set performance, *only* 3 *values were checked* for SH + SG and SG based upon $\sigma = \epsilon \backslash \sqrt{d}$. SH + SG outperforms the competing methods for all 18 permutations of dataset, function and $\varepsilon$.

of the positive and negative eigenvectors of the summands. This is accomplished using an identity expressing the eigenvalues of the summands in terms of $\Sigma^{-1}\delta$ and $\nabla_x f(x_0 + \delta)$, which we state and prove in App. C. The proof is completed by applying the triangle inequality and union bound to combine the two bounds into Eq. (9). Our proof is included in App. C.

# 5 Experiments

## 5.1 Experimental Setup

**Datasets and Models**: Experiments were conducted on a real-world spirometry regression dataset, three image datasets (MNIST [45], FMNIST [90] and CIFAR10 [43]), and one synthetic dataset (Four Quadrant). The Four Quadrant dataset consists of points $x' \in \mathbb{R}^2$ sampled uniformly from the grid $[-2, 2] \times [2, 2] \subseteq \mathbb{R}^2$, with a spacing of 0.008. The label of a point $y(x') = Kx'_1 x'_2 \in \mathbb{R}$ is chosen based upon its quadrant, with $K = 5, 3, 12, -10$ for Quadrants 1, 2, 3 and 4 respectively. We train a 5-layer network on MNIST and FMNIST a ResNet-18 [32] on CIFAR10 and a 6-layer network on Four Quadrant. A 10-layer 1-D convolutional neural network was trained on the spirometry regression dataset. Additional model, hyperparameter and dataset details are outlined in App. E.

**Hardware:** Experiments were performed on an internal cluster using NVIDIA A100 GPUs and AMD EPYC223 7302 16-Core processors.

**Metrics**: Gradient-based explainers are evaluated using a Taylor expansion-like function as a proxy. We estimate a gradient Hessian pair $\tilde{G}_{(x_0,g)} \in \mathbb{R}^d$, $\tilde{H}_{(x_0,g)} \in \mathbb{R}^{d \times d}$ for some function $g : \mathbb{R}^d \to \mathbb{R}$ around point $x_0$. Here $g$ is either a smooth-surrogate of $f$ or $f$ itself. We define the following Taylor expansion-like function:

$$\tilde{f}_{x_0,\tilde{G}_g,\tilde{H}_g}(\Delta) = f(x_0) + \tilde{G}_g^T(\Delta - x_0) + \frac{1}{2}(\Delta - x_0)^T \tilde{H}_g(\Delta - x_0). \tag{10}$$

For readability, we remove the dependence on $x_0$ from $\tilde{G}_{(x_0,g)}$ and $\tilde{H}_{(x_0,g)}$. Eq. (10) is almost equivalent to the Taylor-expansion of $g$ at $x_0$, with the key difference that the zero-th order term is $f(x_0)$ as opposed to $g(x_0)$. This is done to isolate the impact of the explainers $\tilde{G}_g$ and $\tilde{H}_g$. Setting $\tilde{H} = 0$ yields $\tilde{f}_{x_0,\tilde{G}_g,0}(\Delta)$, a first-order Taylor expansion-like function. For brevity, we refer to $\tilde{f}_{x_0,\tilde{G}_g,\tilde{H}_g}$ simply as a Taylor expansion below. We use the following two metrics to quantify the efficacy of SmoothHess:

**Perturbation MSE**: We introduce the Perturbation Mean-Squared-Error ($\mathcal{P}_{MSE}$) to assess the ability of $\tilde{H}$ and/or $\tilde{G}$ to capture the behaviour of $f$ over a given neighborhood. Given point $x_0$ and radius $\varepsilon > 0$, the $\mathcal{P}_{MSE}$ directly measures the fidelity of $\tilde{f}_{x_0,\tilde{G}_g,\tilde{H}_g}$ to $f$ when restricted to the ball $B_\varepsilon(x_0)$:

$$\mathcal{P}_{MSE}(x_0, f, \tilde{f}_{x_0,\tilde{G}_g,\tilde{H}_g}, \varepsilon) = \frac{1}{\text{Vol}(B_\varepsilon(x_0))} \int_{x' \in B_\varepsilon(x_0)} (\tilde{f}_{x_0,\tilde{G}_g,\tilde{H}_g}(x') - f(x'))^2 dx' \tag{11}$$

A low $\mathcal{P}_{MSE}$ value indicates that $\tilde{f}_{x_0,\tilde{G}_g,\tilde{H}_g}$ is a good fit to $f$ when restricted to $B_\varepsilon(x_0)$, and thus that $\tilde{G}_g$ and $\tilde{H}_g$ capture the behaviour of $f$ over $B_\varepsilon(x_0)$. $\mathcal{P}_{MSE}$ may be estimated for a point

| Dataset | MNIST | | | | | FMNIST | | | | | CIFAR10 | | | | |
|---|---|---|---|---|---|---|---|---|---|---|---|---|---|---|---|
| *Attack Magnitude $\epsilon$* | 0.25 | 0.50 | 0.75 | 1.25 | 1.75 | 0.25 | 0.50 | 0.75 | 1.25 | 1.75 | 0.1 | 0.2 | 0.3 | 0.4 | 1.0 |
| SH+SG (Us) | **93.0** | **80.3** | **48.0** | **10.5** | **2.0** | **79.5** | **46.8** | **25.0** | **3.5** | **0.0** | **62.5** | **38.5** | **26.5** | **15.0** | 4.5 |
| SG [76] | 93.3 | 81.8 | 48.8 | 11.3 | 2.8 | **79.5** | 49.3 | 26.3 | 4.0 | **0.0** | 65.0 | 42.0 | 27.5 | 17.0 | **0.0** |
| SP (H + G) | **93.0** | 81.8 | 51.5 | 15.8 | 7.5 | 79.8 | 51.0 | 27.5 | 5.3 | 0.8 | 64.5 | 42.0 | 31.0 | 23.5 | 7.5 |
| SP G | 93.3 | 82.3 | 53.8 | 16.3 | 5.0 | 79.8 | 51.5 | 29.5 | 7.8 | 1.0 | 66.5 | 47.5 | 36.0 | 29.5 | 8.5 |
| G [73] | 93.3 | 82.8 | 56.0 | 18.5 | 8.8 | 80.3 | 52.3 | 31.8 | 11.0 | 2.5 | 69.0 | 51.5 | 41.0 | 34.0 | 21.5 |
| Random | 99.8 | 99.5 | 99.0 | 99.0 | 98.8 | 99.3 | 98.0 | 97.3 | 95.5 | 93.8 | 100.0 | 99.5 | 99.0 | 98.5 | 96.5 |

Table 2: Post-hoc accuracy of adversarial attacks performed on the predicted SoftMax probability, at five attack magnitudes $\epsilon$. Lower is better. Results for SmoothHess + SmoothGrad (SH + SG, Ours), SmoothGrad (SG), SoftPlus Gradient (SP G) and SoftPlus Hessian + SoftPlus Gradient (SP (H + G)) are reported using parameters $\Sigma = \sigma^2 I \in \mathbb{R}^{d \times d}$ and $\beta > 0$ chosen based upon performance on a held-out validation set. We additionally compare with the vanilla (unsmoothed) Gradient (G). First order attack vectors are constructed by scaling the normalized gradient by $\epsilon$ and subtracting from the input. Second order attack vectors are found by minimizing the corresponding second-order Taylor expansions.

$x_0$ by sampling a set of $n \in \mathbb{N}$ points $\{x_i'\}_{i=1}^n$ uniformly from $B_\varepsilon(x_0)$, computing the errors and MC-averaging: $\frac{1}{n} \sum_{i=1}^n (\tilde{f}_{x_0, \tilde{G}_g, \tilde{H}_g}(x_i') - f(x_i'))^2 \approx \mathcal{P}_{MSE}(x_0, f, \tilde{f}_{x_0, \tilde{G}_g, \tilde{H}_g}, \varepsilon)$.

**Adversarial Attacks**: Given $\tilde{G}$ and $\tilde{H}$, one can use $\tilde{f}_{x_0, \tilde{G}, \tilde{H}}$ to construct adversarial attacks of any desired magnitude $\varepsilon > 0$. We denote $\Delta^* \in \mathbb{R}^d$ as the perturbed input, and corresponding attack vectors by $\Delta^* - x_0$. Given only $\tilde{G}_g$ (i.e. $\tilde{H}_g = 0$), the first-order Taylor expansion attack yields: $\Delta^* - x_0 = -\varepsilon \tilde{G}_g \backslash \|\tilde{G}_g\|_2$. Otherwise, the attack may be framed as the solution to a quadratic optimization minimizing the second-order Taylor expansion:

$$\min_{\Delta \in \mathbb{R}^d} \quad \tilde{f}_{x_0, \tilde{G}_g, \tilde{H}_g}(\Delta), \quad s.t. \quad \|\Delta - x_0\|_2 \leq \varepsilon. \tag{12}$$

Although this optimization is non-convex (as $\tilde{H}_g$ is not guaranteed to be positive-semi-definite), it can be solved exactly [12]. For implementation details, and a discussion of the similarities and differences with Singla et al. [74], see App. D. We set $g$ to be the smoothed version of the predicted class SoftMax probability function. Given a set of test points $\{x_i\}_{i=1}^n, n \in \mathbb{N}$, we validate the efficacy of our attacks using the average post-hoc accuracy metric $\frac{1}{n} \sum_{i=1}^n \mathbb{I}[\arg\max_t F(x_i) = \arg\max_t F(\Delta_i^*)]$, where $\mathbb{I}$ denotes the indicator function, and $\Delta_i^*$ the output after $x_i$ is attacked.

**Setup**: We provide the details for our experiments below:

**Four Quadrant:** We train the network to memorize the Four Quadrant dataset. We measure the interactions at $x_0 = (0, 0)^T$ using both SmoothHess and SoftPlus Hessian, estimated with granularly sampled $\sigma^2 \in \{1e\text{-}3, \dots, 1\}$ and $\beta \in \{1e\text{-}1, \dots, 4\}$.

$\mathcal{P}_{MSE}$ **:** We set $g$ to be the smoothed version of a predicted class logit or a penultimate neuron. The penultimate neuron is chosen to be the maximally activated neuron on average over the train data by the "three," dress and cat classes for MNIST, FMNIST and CIFAR10, respectively. We compute $\mathcal{P}_{MSE}$ for three neighborhood sizes $\varepsilon \in \{0.25, 0.50, 1.0\}$. While *over one-hundred values* of $\beta$ are checked on a validation set before selection, *only three values* of $\sigma$ are checked for SmoothHess and SmoothGrad, based on the common-sense criterion $\sigma = \varepsilon/\sqrt{d}$ : $\sigma \in \{\varepsilon/2\sqrt{d}, 3\varepsilon/4\sqrt{d}, \varepsilon/\sqrt{d}\}$ outlined in Sec 4. The standard deviation of $\mathcal{P}_{MSE}$ results is reported in App. F.

**Adversarial Attacks:** Adversarial attacks are performed after selecting the best $\beta$ and $\sigma$ from a held-out validation set. We refrain from using the criterion $\sigma = \varepsilon/\sqrt{d}$ for adversarial attacks, as they rely upon the extremal, as opposed to average, behavior of $f$. We check between $\approx 10$ and $\approx 30$ values of $\sigma$ and $\beta$ on the held-out validation set before selecting the values resulting in the most effective attacks. Attacks are performed using magnitudes $\varepsilon \in \{0.25, 0.50, 0.75, 1.25, 1.75\}$ for MNIST and FMNIST and $\varepsilon \in \{0.1, 0.2, 0.3, 0.4, 1.0\}$ for CIFAR10, which is easier to successfully attack at lower magnitudes due to it's complexity. For more details see App. D.

**Competing Methods**: We compare SmoothHess with other gradient-based methods that model $f$ locally; i.e. those which can be associated with the Taylor expansion around a smooth surrogate of $f$ or $f$ itself. Specifically, we compare with the Gradient and Hessian of SoftPlus smoothed network $f_\beta$ [20, 39], SmoothGrad [76] and the vanilla (unsmoothed) Gradient. We also compare adversarial attacks with random vectors scaled to the attack magnitude $\varepsilon$ as a baseline.

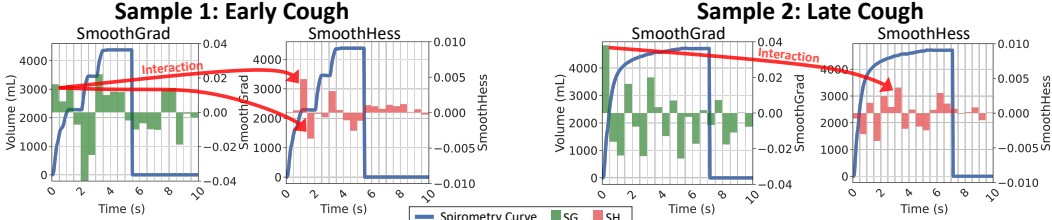

Figure 3: Evaluation of network predictions for two spirometry samples (Left and Right), using SmoothGrad and SmoothHess. Spirometry curves are plotted in blue. SmoothGrad (green) and SmoothHess (red) attributions are grouped into 0.5s intervals and plotted as bars for each interval. The plotted SmoothHess values are interactions with respect to the first 0.5s time interval. The red arrows point to the features that exhibit the largest interactions (endpoint) with the initial 0.5s (source). Sample 1 exhibits coughing within the first 4 seconds, as indicated by plateauing spirometry curve. We observe that the strongest interactions for Sample 1 occur at this initial cough-induced plateau. In contrast, Sample 2 exhibits no coughing within the first 4 seconds; the strongest interactions occur later near small fluctuations in the spirometry curve.

Additional experiments are provided in App. F. We compare the efficacy of SmoothHess with the unsmoothed Hessian (for adversarial attacks on SoftMax) and Swish [65] smoothed networks, both of which our method outperforms. We also run the $\mathcal{P}_{MSE}$ experiment using a ResNet101 model, validating the superior ability of SmoothHess to capture interactions in larger networks.

## 5.2 Results

**Symmetric Smoothing - Four Quadrant Dataset:** We investigate the ability of both SmoothHess and the SoftPlus Hessian to capture local feature interactions symmetrically. Results are shown in Figure 2. At each value of $\sigma^2$, aside from extremely small $\sigma^2 < 1e$-2.5, the off-diagonal element of SmoothHess is approximately equal to 2.5, the average interaction over the quadrants. The off-diagonal element of the SoftPlus Hessian is essentially never equal to 2.5. The results for SmoothHess follow from the fact that an isotropic covariance was used, the rotation invariance of which weights points that are equidistant from $x$ equally. The SmoothHess off-diagonal element is not $\approx 2.5$ at small $\sigma$ because, despite the network being trained to memorize the data, such a small neighborhood will inherently reflect noise. The inability of SoftPlus to capture interactions symmetrically follows from the fact that smoothing is done internally on each neuron, which does not guarantee symmetry.

**Perturbation Mean-Squared-Error:** We show SmoothHess can be used to capture interactions occurring in a variety of neighborhoods around $x$. It is shown in Table 1 that SmoothHess + SmoothGrad (SH + SG) achieves the lowest $\mathcal{P}_{MSE}$ for all 18 combinations of dataset, function and neighborhood size $\varepsilon$. We emphasize that this is despite the fact that *only three values* of $\sigma$ were validated for SG + SH and SG on a held-out set compared to *over one-hundred values* of $\beta$ for SP (H + G) and SP G. This indicates the superior ability of SmoothHess to model $f$ over a diverse set of neighborhood sizes $\varepsilon$. Further, it can be seen that second-order methods SH + SG and SP (H + G) achieve significantly lower $\mathcal{P}_{MSE}$ than their first-order counterparts, SG and SP G, respectively, sometimes by an order of magnitude. This confirms the intuition that higher order Taylor expansions of the smooth surrogate provide more accurate models of $f$. Interestingly, while SH + SG is clearly superior to SP (H + G), we see that SG and SP G are tied in many cases. This could indicate that the symmetric properties of Gaussian smoothing are comparatively more important when higher-order derivatives are considered. However, more investigation is needed.

**Adversarial Attacks:** We use the interactions found by SmoothHess to generate adversarial attacks on the classifier. We see from Table 2 that the Taylor expansion associated with SmoothHess and SmoothGrad generates the most powerful adversarial attacks for all datasets and attack magnitudes $\varepsilon$, aside from CIFAR10 at $\varepsilon = 1.0$ where both methods successfully change most class predictions. The superiority of the SmoothHess attacks indicates that the Gaussian smoothing is capturing the extremal behavior of the predicted SoftMax probability more effectively than the other methods. One hypothesis for SmoothGrad outperforming SmoothHess in the largest neighborhood for CIFAR10 is that the network behavior is highly complex over a large area, and adding higher order terms decreases performance.

**Qualitative Analysis of FEV$_1$ Prediction using Rejected Spirometry:**

A spirometry test is a common procedure used to evaluate pulmonary function, and is the main diagnostic tool for lung diseases such as Chronic Obstructive Pulmonary Disease [40]. During an exam, the patient blows into a spirometer, which records the exhalation as a volume-time curve. Recent works have investigated the use of deep learning on spirograms for tasks such as subtyping [11], genetics [16, 91], and mortality prediction [36]. Additional background on spirometry is outlined in App. E.

Traditionally, exhalations interrupted by coughing are discarded for quality control reasons. We train a CNN on raw spirograms from the UK Biobank [79] to predict the patient's "Forced Expiratory Volume in 1 Second" (FEV$_1$), a metric frequently used to evaluate lung health, using efforts that were rejected due to coughing. In Fig. 3, we apply SmoothHess and SmoothGrad on two spirometry samples to understand their FEV$_1$ predictions. In Sample 1, coughing occurs within the first 4 seconds of the curve, as evidenced by the early plateauing of the curve which indicates a pause in exhalation. In contrast, Sample



Figure 4: Heatmap of the SmoothHess interactions for the spirometry sample in Fig. 3, calculated on CNNs with varying convolution kernel width.

2 exhibits no detectable coughing within the first 4 seconds. To improve the interpretability of the results, we group the features into 0.5 second time intervals. FEV$_1$ is traditionally measured in the initial 2 seconds of the non-rejected samples (see App. E), therefore we calculate SmoothHess interactions with respect to the first 0.5 second time interval.

FEV$_1$ is known to be strongly affected by the presence of coughing [55]. The time intervals where coughing occurs, indicated by plateaus in the spirometry curves in Figure 3, should be an important signal for any model trained to predict FEV$_1$. Indeed, we observe that for Sample 1, the SmoothGrad attribution for the first 0.5s interval is relatively low, with strong interactions occurring at the cough-induced plateaus. In contrast, the first 0.5s interval for Sample 2 shows high importance, with lower magnitude interactions that may be indicative of small fluctuations in the volume-time curve.

In Figure 4 we present the SmoothHess matrix for Sample 1 in Fig. 3, applied on two CNN models with different convolution kernel width. Interestingly, the smaller kernel width constrains the interactions to features that are spatially close together. In contrast, the features in the large kernel model have long-ranging interactions. These results agree with the intuition that smaller kernel widths, which constrain the receptive field of neurons up until the final layers, may correspondingly limit feature interactions.

## 6    Conclusion and Future Work

We introduce SmoothHess, the Hessian of the network convolved with a Gaussian, as a method for quantifying second-order feature interactions. SmoothHess estimation, which relies only on gradient oracle calls, and cannot be performed with naive MC-averaging, is made possible by an extension of Stein's Lemma that we derive. We provide non-asymptotic complexity bounds for our estimation procedure. In contrast to previous works, our method can be run post-hoc, does not require architecture changes, affords the user localized flexibility over the weighting of input space, and can be run on any network output. We experimentally validate the superior ability of SmoothHess to capture interactions over a variety of neighborhoods compared to competing methods. Last, we use SmoothHess to glean insight into a network trained on real-world spirometry data.

**Limitations:** The outer product computations in SmoothHess estimation have space and time complexity $\mathcal{O}(d^2)$. When $d \gg 0$ this becomes expensive: for instance ImageNet [19] typically has $d^2 \approx 10^{10}$. This is a common problem for all interaction effect methods. We leave as future work to explore computationally efficient alternatives or approximations to these outer products; a potential remedy is to devise an appropriate power method to estimate the top eigenvectors of SmoothHess, rather than its entirety, which fits well our estimation via sampling low-rank matrices.

## Acknowledgements

This project was supported by NIH grants R01CA240771 and U24CA264369 from NCI, in part by MSKCC's Cancer Center core support NIH grant P30CA008748 from NCI, and NIH 2T32HL007427-41.

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

## A  Societal Impacts

Machine learning models have become increasingly pervasive in society, from medicine [10, 67, 15] and law [38] to entertainment[14, 9, 80]. Therefore, it is important that users of the technology understand the factors underlying model predictions. To this end we propose SmoothHess for quantifying the feature interactions affecting model output. Potential applications of our method are widespread; SmoothHess may be used to find interactions influencing a model to predict whether a customer will default on a credit loan or if a patient has melanoma. The deeper understanding of the model gleaned from SmoothHess may be used to improve decision making. For instance, a doctor may notice that the SmoothHess feature interactions between the pixels in an image of a lesion don't "make sense", indicating that the model should not be trusted and that a more granular human assessment is required. However, such applications are highly sensitive and the cost of inaccurate predictions, or, in this case, misinterpreted attributions, can be high. An inaccurate interpretation can instill unwarranted trust, or mistrust, in a user. In the most extreme cases this may lead to sub-optimal decision making such as the confident denial of a credit loan to a trustworthy customer or misdiagnosis of a benign lesion as malignant.

In addition, it is important to note the challenges related to ensuring robust, stable, and trustworthy explanations. In particular, recent works have uncovered issues related to the sensitivity of the explainer to small changes in the input [5, 41], adversarial attacks [27, 20, 37], or hyperparameter tuning [8]. Methods have been proposed that attempt to quantify explanation uncertainty [75, 69, 35], however further challenges remain. Thus, as with all methods for explaining machine learning model predictions, we recommend that SmoothHess is used in tandem with the careful consideration of domain experts, who are best equipped to interpret interactions in the context of their field.

## B  Proof of Proposition 1

### B.1  Preliminary

We make use of a Lemma from Lin et al. [49] in our proof below, which we relate here:

**Lemma 3.** *(Lin et al. [49]) Denote $x_0 \in \mathbb{R}^d$, locally-Lipschitz continuous function $h(z) : \mathbb{R}^d \to \mathbb{R}$, covariance matrix $\Sigma \in \mathbb{R}^{d \times d}$, random vector $z \in \mathbb{R}^d$ distributed from $z \sim \mathcal{N}(x_0, \Sigma)$. If $\mathbb{E}_z[|h(z)|] < \infty$, then*

$$\mathbb{E}_z[\Sigma^{-1}((z - x_0)(z - x_0)^T - \Sigma)\Sigma^{-1}h(z)] = \mathbb{E}_z[\Sigma^{-1}(z - x_0)[\nabla_z h(z)]^T]. \tag{13}$$

### B.2  Main Result

**Proposition 1.** *Given $x_0 \in \mathbb{R}^d$, $L$-Lipschitz continuous function $g : \mathbb{R}^d \to \mathbb{R}$, covariance matrix $\Sigma \in \mathbb{R}^{d \times d}$ and random vector $\delta \in \mathbb{R}^d$ distributed from $\delta \sim \mathcal{N}(0, \Sigma)$ with density function $q_\Sigma : \mathbb{R}^d \to \mathbb{R}$, then*

$$\mathbb{E}_\delta[\Sigma^{-1}\delta[\nabla_x g(x_0 + \delta)]^T] = \nabla_x^2[(g * q_\Sigma)(x_0)] = \nabla_x^2 h_{g,\Sigma}(x_0), \tag{7}$$

*where $*$ denotes convolution.*

*Proof.* We define random vector $z = x_0 + \delta \in \mathbb{R}^d$ distributed from $z \sim \mathcal{N}(x_0, \Sigma)$, and for which we denote the density function as $p_\Sigma$. We begin by showing $\mathbb{E}_z[|g(z)|] = \mathbb{E}_\delta[|g(x_0 + \delta)|] < \infty$. We are given that $g$ is $L$-Lipschitz, i.e. $\forall a, b \in \mathbb{R}^d$

$$|g(a) - g(b)| \leq L\|a - b\|_2 \tag{14}$$

for some $L > 0$.

Now, for any fixed $\delta \in \mathbb{R}^d$ we have

$$|g(x_0 + \delta)| = |g(x_0) + (g(x_0 + \delta) - g(x_0))| \stackrel{\triangle \text{ ineq.}}{\leq} |g(x_0)| + |g(x_0 + \delta) - g(x_0)| \stackrel{Eq.14}{\leq} \tag{15a}$$

$$|g(x_0)| + L\|x_0 + \delta - x_0\|_2 = |g(x_0)| + L\|\delta\|_2. \tag{15b}$$

Thus, we may bound the expectation $\mathbb{E}_\delta[|g(x_0 + \delta)|]$:

$$\mathbb{E}_\delta[|g(x_0 + \delta)|] \leq \mathbb{E}_\delta[|g(x_0)| + L\|\delta\|_2] = g(x_0) + L\mathbb{E}_\delta[\|\delta\|_2] < \infty. \tag{16}$$

Here $g(x_0) < \infty$ as it is a constant, and it may be seen that $L\mathbb{E}_\delta[\|\delta\|_2] < \infty$ using a simple change of variables. Defining $\beta = \Sigma^{-\frac{1}{2}}\delta \sim \mathcal{N}(0, I)$ one may write $\mathbb{E}_\delta[\|\delta\|_2] = \mathbb{E}_\beta[\|\Sigma^{\frac{1}{2}}\beta\|_2]$. As a straightforward consequence of Cauchy-Schwarz, for any fixed $\beta$, one may write

$$\|\Sigma^{\frac{1}{2}}\beta\|_2 \leq \|\Sigma^{\frac{1}{2}}\|_F\|\beta\|_2 \tag{17}$$

where $\|\cdot\|_F$ denotes the Frobenius norm. Noting that $\|\beta\|_2 \sim \mathcal{X}_d$ we use Eq. (17) to see

$$\mathbb{E}_\beta[\|\Sigma^{\frac{1}{2}}\beta\|_2] \leq \mathbb{E}_\beta[\|\Sigma^{\frac{1}{2}}\|_F\|\beta\|_2] = \|\Sigma^{\frac{1}{2}}\|_F\mathbb{E}_\beta[\|\beta\|_2] = \|\Sigma^{\frac{1}{2}}\|_F\sqrt{2}\frac{\Gamma((d+1)/2)}{\Gamma(d/2)} < \infty \tag{18}$$

where $\Gamma(\cdot)$ denotes the Gamma function.

Next, we move the Hessian operator inside the integral.

$$\nabla_x^2(g * q_\Sigma)(x_0) = \nabla_x^2 \int_{z \in \mathbb{R}^d} g(z)q_\Sigma(z - x_0)dz = \int_{z \in \mathbb{R}^d} g(z)\nabla_x^2 q_\Sigma(z - x_0)dz = \tag{19a}$$

$$\int_{z \in \mathbb{R}^d} g(z)\nabla_x\nabla_x^T q_\Sigma(z - x_0)dz = \int_{z \in \mathbb{R}^d} g(z)(\nabla_x q_\Sigma(z - x_0)(z - x_0)^T\Sigma^{-1})dz = \tag{19b}$$

$$\int_{z \in \mathbb{R}^d} g(z)((\nabla_x(z - x_0)^T\Sigma^{-1})q_\Sigma(z - x_0) + (\nabla_x q_\Sigma(z - x_0))(z - x_0)^T\Sigma^{-1})dz = \tag{19c}$$

$$\int_{z \in \mathbb{R}^d} g(z)(-I\Sigma^{-1}q_\Sigma(z - x_0) + (\nabla_x q_\Sigma(z - x_0))(z - x_0)^T\Sigma^{-1})dz = \tag{19d}$$

$$\int_{z \in \mathbb{R}^d} g(z)(-I\Sigma^{-1}q_\Sigma(z - x_0) + q_\Sigma(z - x_0)\Sigma^{-1}(z - x_0)(z - x_0)^T\Sigma^{-1})dz = \tag{19e}$$

$$\int_{z \in \mathbb{R}^d} g(z)q_\Sigma(z - x_0)(-I\Sigma^{-1} + \Sigma^{-1}(z - x_0)(z - x_0)^T\Sigma^{-1})dz = \tag{19f}$$

$$\int_{z \in \mathbb{R}^d} g(z)p_\Sigma(z)(-I\Sigma^{-1} + \Sigma^{-1}(z - x_0)(z - x_0)^T\Sigma^{-1})dz = \quad (p_\Sigma(z) = q_\Sigma(z - x_0)) \tag{19g}$$

$$\mathbb{E}_z[g(z)(-I\Sigma^{-1} + \Sigma^{-1}(z - x_0)(z - x_0)^T\Sigma^{-1})] = \tag{19h}$$

$$\mathbb{E}_z[g(z)\Sigma^{-1}(-I + (z - x_0)(z - x_0)^T\Sigma^{-1})] = \tag{19i}$$

$$\mathbb{E}_z[g(z)\Sigma^{-1}(-\Sigma + (z - x_0)(z - x_0)^T)\Sigma^{-1}] = \tag{19j}$$

$$\mathbb{E}_z[\Sigma^{-1}((z - x_0)(z - x_0)^T - \Sigma)\Sigma^{-1}g(z)] \tag{19k}$$

Lemma 3 may be applied as $g$ is Lipschitz, and thus locally-Lipschitz, and $\mathbb{E}_z[|g(z)|] < \infty$, yielding

$$\mathbb{E}_z[\Sigma^{-1}((z - x_0)(z - x_0)^T - \Sigma)\Sigma^{-1}g(z)] = \mathbb{E}_z[\Sigma^{-1}(z - x_0)[\nabla_z g(z)]^T] \tag{20}$$

Using a change of variables from $z$ to $x_0 + \delta$ we write

$$\mathbb{E}_z[\Sigma^{-1}(z - x_0)[\nabla_z g(z)]^T] = \mathbb{E}_\delta[\Sigma^{-1}\delta[\nabla_x g(x_0 + \delta)]^T] \tag{21}$$

which, when combined with Eq. (19) and Eq. (20), completes the proof

$$\nabla_x^2(g * q_\Sigma)(x_0) = \mathbb{E}_\delta[\Sigma^{-1}\delta[\nabla_x g(x_0 + \delta)]^T]. \tag{22}$$

$\square$

## C   Proof of Theorem 1

We begin by establishing a result expressing the eigenvalues of the symmetrization of a rank 1 matrix in closed form:

**Lemma 4.** *Given $x, y \in \mathbb{R}^d$ denote $A = xy^T + yx^T \in \mathbb{R}^{d \times d}$. The following facts hold:*

    *1. Matrix A will have the following eigenvalues*

        • *It has $d - 2$ eigenvalues equal to $0$*

- *The other two eigenvalues are denoted by $\lambda^+(A)$ and $\lambda^-(A)$ will have the following form:*

$$\lambda^\pm(A) = x^T y \pm \|x\|_2 \|y\|_2 \tag{23}$$

2. *$\lambda^+(A)$ and $\lambda^-(A)$ are non-negative and non-positive respectively.*

3. *Given $x$ and $y$ are sampled from sub-gaussian distributions, then $\lambda^+(A)$ and $\lambda^-(A)$ are sub-exponential random variables.*

*Proof.* We divide the proof into three sections

1. We have $\text{rank}(A) \le 2$, $A \in S_d$. Therefore $\exists Q = [e_1|e_2|\ldots|e_d] \in \mathbb{R}^{d \times d}$ s.t. $Q^T Q = QQ^T = I$, the column vectors $e_i \in \mathbb{R}^d$ are orthonormal and

$$A = Q\Lambda Q^T \tag{24a}$$

$$Q^T A Q = \Lambda \tag{24b}$$

where $\Lambda = \text{diag}([\lambda_1, \lambda_2, 0, \ldots, 0]) \in \mathbb{R}^{d \times d}$. As eigenvalues are invariant to change of basis, $A$ has eigenvalues $\lambda_1$ and $\lambda_2$ and the other $d - 2$ eigenvalues are equal to $0$.

It can be seen that $\text{span}(\{x, y\}) = \text{span}(\{e_1, e_2\})$. As $\text{span}(\{e_1, e_2\}) = C(A)$ this can be shown by proving $\text{span}(\{x, y\}) = C(A)$. We know $\exists z_x \in \mathbb{R}^d : z_x \perp x$ and $\exists z_y \in \mathbb{R}^d : z_y \perp y$. Thus we have

$$A z_x = (xy^T + yx^T) z_x = xy^T z_x + yx^T z_x = x(y^T z_x) \tag{25a}$$

$$A z_y = (xy^T + yx^T) z_y = xy^T z_y + yx^T z_y = y(x^T z_y). \tag{25b}$$

From the above, we see that $x, y \in C(A)$. We know $\text{rank}(A) \le 2$. If $\text{rank}(A) = 2$ we have $x \ne y$ and thus $\text{span}(\{x, y\}) = C(A)$. If $\text{rank}(A) = 1$ we have $x \ne 0, y \ne 0$ and still $\text{span}(\{x, y\}) = C(A)$. If $\text{rank}(A) = 0$ we have $x = y = 0$ and clearly $\text{span}(\{x, y\}) = \{0\} = C(A)$.

As $\{e_i\}_{i=1}^d$ are orthonormal, we have that $x^T e_j = y^T e_j = 0$, $\forall j > 2$. We define

$$\bar{x} = Q^T x = (e_1^T x, e_2^T x, e_3^T x, \ldots, e_d^T x) = (e_1^T x, e_2^T x, 0, \ldots, 0) \in \mathbb{R}^d, \tag{26a}$$

$$\bar{y} = Q^T y = (e_1^T y, e_2^T y, e_3^T y, \ldots, e_d^T y) = (e_1^T y, e_2^T y, 0, \ldots, 0) \in \mathbb{R}^d \tag{26b}$$

We define $Q_2 = [e_1|e_2] \in \mathbb{R}^{d \times 2}$ and

$$\tilde{x} = Q_2^T x = (e_1^T x, e_2^T x) \in \mathbb{R}^2 \tag{27a}$$

$$\tilde{y} = Q_2^T y = (e_1^T y, e_2^T y) \in \mathbb{R}^2 \tag{27b}$$

We first show that $\|\bar{x}\|_2$ and $\|\bar{y}\|_2$ are equal to $\|x\|_2$ and $\|y\|_2$ respectively:

$$\|\bar{x}\|_2 = \|Q^T x\|_2 = \sqrt{(Q^T x)^T (Q^T x)} = \sqrt{x^T Q Q^T x} = \sqrt{x^T I x} = \|x\|_2 \tag{28a}$$

$$\|\bar{y}\|_2 = \|Q^T y\|_2 = \sqrt{(Q^T y)^T (Q^T y)} = \sqrt{y^T Q Q^T y} = \sqrt{y^T I y} = \|y\|_2 \tag{28b}$$

Next, we use Eq. (28) to show that $\|\tilde{x}\|_2$ and $\|\tilde{y}\|_2$ are equal to $\|x\|_2$ and $\|y\|_2$ respectively:

$$\|\tilde{x}\|_2 = \sqrt{(e_1^T x)^2 + (e_2^T x)^2} = \|\bar{x}\|_2 = \|x\|_2 \tag{29a}$$

$$\|\tilde{y}\|_2 = \sqrt{(e_1^T y)^2 + (e_2^T y)^2} = \|\bar{y}\|_2 = \|y\|_2 \tag{29b}$$

We show an equality between inner products $\tilde{x}^T \tilde{y} = x^T y$:

$$\tilde{x}^T \tilde{y} = (e_1^T x)(e_1^T y) + (e_2^T x)(e_2^T y) = \bar{x}^T \bar{y} = (Q^T x)^T Q^T y = x^T Q Q^T y = \tag{30a}$$

$$x^T I y = x^T y \tag{30b}$$

Now, one may write,

$$Q_2^T A Q_2 = \text{diag}([\lambda_1, \lambda_2]) \tag{31a}$$

$$Q_2^T (xy^T + yx^T) Q_2 = \text{diag}([\lambda_1, \lambda_2]) \tag{31b}$$

$$Q_2^T xy^T Q_2 + Q_2^T yx^T Q_2 = \text{diag}([\lambda_1, \lambda_2]) \tag{31c}$$

$$\tilde{x}\tilde{y}^T + \tilde{y}\tilde{x}^T = \text{diag}([\lambda_1, \lambda_2]). \tag{31d}$$

The following facts hold as a result of Eq. (31d):

- $\lambda_1 = 2\tilde{x}_1\tilde{y}_1$
- $\lambda_2 = 2\tilde{x}_2\tilde{y}_2$
- $\tilde{x}_1\tilde{y}_2 + \tilde{x}_2\tilde{y}_1 = 0$

Now, we show

$$\lambda_1 = \tilde{x}^T\tilde{y} + \|\tilde{x}\|_2\|\tilde{y}\|_2, \qquad \lambda_2 = \tilde{x}^T\tilde{y} - \|\tilde{x}\|_2\|\tilde{y}\|_2 \tag{32}$$

Which can be derived as such:

$$\tilde{x}^T\tilde{y} \pm \|\tilde{x}\|_2\|\tilde{y}\|_2 = \tilde{x}_1\tilde{y}_1 + \tilde{x}_2\tilde{y}_2 \pm \sqrt{(\tilde{x}_1^2 + \tilde{x}_2^2)(\tilde{y}_1^2 + \tilde{y}_2^2)} \tag{33}$$

$$= \tilde{x}_1\tilde{y}_1 + \tilde{x}_2\tilde{y}_2 \pm \sqrt{(\tilde{x}_1\tilde{y}_1 - \tilde{x}_2\tilde{y}_2)^2} = \tilde{x}_1\tilde{y}_1 + \tilde{x}_2\tilde{y}_2 \pm |\tilde{x}_1\tilde{y}_1 - \tilde{x}_2\tilde{y}_2| = \lambda_1 \text{ or } \lambda_2$$

where the second equality comes from the following:

$$(\tilde{x}_1\tilde{y}_1 - \tilde{x}_2\tilde{y}_2)^2 = (\tilde{x}_1\tilde{y}_1)^2 - 2\tilde{x}_1\tilde{x}_2\tilde{y}_1\tilde{y}_2 + (\tilde{x}_2\tilde{y}_2)^2$$

$$= (\tilde{x}_1\tilde{y}_1)^2 - 2\tilde{x}_1\tilde{x}_2\tilde{y}_1\tilde{y}_2 + (\tilde{x}_2\tilde{y}_2)^2 + (\tilde{x}_1\tilde{y}_2 + \tilde{x}_2\tilde{y}_1)^2 \tag{34}$$

$$= (\tilde{x_1}^2 + \tilde{x}_2^2)(\tilde{y}_1^2 + \tilde{y}_2^2).$$

Proving that Eq. (32) holds. Finally, we combine Eq. (29) and Eq. (30) with Eq. (32) to express the eigenvalues of $A$ as:

$$\lambda_1 = x^T y + \|x\|_2\|y\|_2, \qquad \lambda_2 = x^T y - \|x\|_2\|y\|_2 \tag{35}$$

which we use to denote $\lambda^+(A) = \lambda_1, \lambda^-(A) = \lambda_2$.

2. As $|x^T y| \leq \|x\|_2\|y\|_2$, it follows that $\lambda^+(A) = x^T y + \|x\|_2\|y\|_2 \geq 0$ and $\lambda^-(A) = x^T y - \|x\|_2\|y\|_2 \leq 0$.

3. We denote $D = \{1, \ldots, d\}$ It can be seen that $x_i y_i$ is sub-exponential $\forall i \in D$, as a sub-gaussian times a sub-gaussian is sub exponential. Thus, it follows that

$$x^T y = \sum_{i=1}^{d} x_i y_i \text{ is a sub-exponential random variable,} \tag{36}$$

as the sum of sub-exponential random variables is sub-exponential. Further we see that $x_i^2$ and $y_i^2$ are sub-exponential as the square of a sub-gaussian is sub-exponential. As the sum of sub-exponentials is sub-exponential we have that $\sum_{i=1}^{d} x_i^2, \sum_{i=1}^{d} y_i^2$ are both sub-exponential random variables. As the square root of a sub-exponential is sub-gaussian we have that

$$\|x\|_2 = \sqrt{\sum_{i=1}^{d} x_i^2} \text{ is a sub-gaussian random variable} \tag{37a}$$

$$\|y\|_2 = \sqrt{\sum_{i=1}^{d} y_i^2} \text{ is a sub-gaussian random variable} \tag{37b}$$

As a sub-gaussian times a sub-gaussian is sub-exponential, from Eq. 37 we have that

$$\|x\|_2\|y\|_2 \text{ is a sub-exponential random variable .} \tag{38}$$

Now we see from Eq. and 36 Eq. 38 that both $\lambda^+(A)$ and $\lambda^-(A)$ are the sum of sub-exponential random variables and thus are sub-exponential.

$\square$

We now use the result of Lemma 4 to prove the sample complexity bounds for SmoothHess in Theorem 1:

**Theorem 1.** *Let* $f : \mathbb{R}^d \to \mathbb{R}$ *be a piece-wise linear function over a finite partition of* $\mathbb{R}^d$. *Let* $x_0 \in \mathbb{R}^d$, *and denote* $\{\delta_i\}_{i=1}^n$, *a set of* $n$ *i.i.d random vectors in* $\mathbb{R}^d$ *distributed from* $\delta_i \sim \mathcal{N}(0, \Sigma)$. *Given* $\hat{H}_n(x_0, f, \Sigma)$ *as in Eq.* (8), *for any fixed* $\varepsilon, \gamma \in (0, 1]$, *given* $n \geq \frac{4}{\varepsilon^2}[\max((C^+\sqrt{d} + \sqrt{\frac{1}{c^+}\log\frac{4}{\gamma}})^2, (C^-\sqrt{d} + \sqrt{\frac{1}{c^-}\log\frac{4}{\gamma}})^2)]$ *then*

$$\mathbb{P}\left(\left\|\hat{H}_n - H\right\|_2 > \varepsilon\right) \leq \gamma, \tag{9}$$

*where* $H = \nabla_x^2[(f * q_\Sigma)(x_0)]$, $C^+, C^- c^+, c^- > 0$ *are constants depending on the function* $f$ *and covariance* $\Sigma$ *and* $q_\Sigma : \mathbb{R}^d \to \mathbb{R}$ *is the density function of* $\mathcal{N}(0, \Sigma)$.

*Proof.* As $x_0, f$ and $\Sigma$ are fixed, we refer to $\hat{H}_n(f, x_0, \Sigma)$ as $\hat{H}_n$ for brevity. We denote $D = \{1, \ldots, d\}$. We begin by explicitly expressing our estimator $\hat{H}_n$ in terms of $\delta_i$ and $\nabla_x f(x_0 + \delta_i)$. From Eq. (8) we have

$$H_n^\circ = \frac{1}{n}\sum_{i=1}^n \Sigma^{-1}\delta_i[\nabla_x f(x_0 + \delta_i)]^T \tag{39a}$$

$$\hat{H}_n = \frac{1}{2}H_n^\circ + \frac{1}{2}H_n^{\circ T} = \tag{39b}$$

$$\frac{1}{2}\frac{1}{n}\sum_{i=1}^n(\Sigma^{-1}\delta_i[\nabla_x f(x_0 + \delta_i)]^T) + \frac{1}{2}\frac{1}{n}\sum_{i=1}^n(\nabla_x f(x_0 + \delta_i)\delta_i^T \Sigma^{-1}) \tag{39c}$$

Now, we show the convergence of our estimator $\hat{H}_n$:

**Lemma 5.** $\lim_{n\to\infty} \hat{H}_n = H$

*Proof.* From Proposition 1 it is clear to see that $\lim_{n\to\infty} H_n^\circ = H$:

$$\lim_{n\to\infty} H_n^\circ = \lim_{n\to\infty}\frac{1}{n}\sum_{i=1}^n \Sigma^{-1}\delta_i[\nabla_x f(x_0 + \delta_i)]^T = \mathbb{E}_\delta[\Sigma^{-1}\delta[\nabla_x f(x_0 + \delta)]^T] \overset{\text{Prop } 1}{=} H. \tag{40}$$

Next, we show it is also the case that $\lim_{n\to\infty} H_n^{\circ T} = H$:

$$\lim_{n\to\infty} H_n^{\circ T} = \lim_{n\to\infty}\frac{1}{n}\sum_{i=1}^n \nabla_x f(x_0 + \delta_i)\delta_i^T \Sigma^{-1} = \tag{41a}$$

$$\lim_{n\to\infty}\frac{1}{n}\sum_{i=1}^n(\Sigma^{-1}\delta_i[\nabla_x f(x_0 + \delta_i)]^T)^T = (\lim_{n\to\infty}\frac{1}{n}\sum_{i=1}^n \Sigma^{-1}\delta_i[\nabla_x f(x_0 + \delta_i)]^T)^T = \tag{41b}$$

$$(\mathbb{E}_\delta[\Sigma^{-1}\delta[\nabla_x f(x_0 + \delta)]^T])^T \overset{\text{Prop } 1}{=} H^T = H \qquad \text{(Symmetry of Hessian)} \tag{41c}$$

Now, as we have $\lim_{n\to\infty} H_n^\circ = \lim_{n\to\infty} H_n^{\circ T} = H$, it follows that

$$\lim_{n\to\infty} \hat{H}_n = \lim_{n\to\infty}\frac{1}{2}H_n^\circ + \lim_{n\to\infty}\frac{1}{2}H_n^{\circ T} = \frac{1}{2}H + \frac{1}{2}H = H \tag{42}$$

$\square$

We establish the following notation to be used below: given a fixed vector $\delta \in \mathbb{R}^d$ one may construct matrix $A_\delta \in \mathbb{R}^{d\times d}$ by:

$$A_\delta = \frac{1}{2}(\Sigma^{-1}\delta[\nabla_x f(x_0 + \delta)]^T) + \frac{1}{2}(\nabla_x f(x_0 + \delta)\delta^T \Sigma^{-1}) \in \mathbb{R}^{d\times d} \tag{43}$$

It can be seen from Eq. 39 and Lemma 5 that $H$ and $\hat{H}_n$ may be expressed in terms of matrices $A_\delta$ and $A_{\delta_i}$:

$$H = \mathbb{E}_\delta[A_\delta], \qquad\qquad \hat{H}_n = \frac{1}{n}\sum_{i=1}^n A_{\delta_i} \qquad (44)$$

Next, we establish that the random vectors $\Sigma^{-1}\delta$ and $\nabla f(x_0 + \delta)$ are sub-gaussian:

**Lemma 6.** *The random vectors $\Sigma^{-1}\delta$ and $\nabla f(x_0 + \delta)$ are sub-gaussian.*

*Proof.* As $\Sigma^{-1}\delta$ is Gaussian it is sub-gaussian. Now, we show that $\nabla f(x_0 + \delta)$ is a sub-gaussian random-vector. We have that $f$ is piecewise-linear over a partition of $\mathbb{R}^d$ with finite cardinality $L$. Let us denote this partition as $\mathcal{Q} = \{Q_i\}_{i=1}^L$, $Q_i \subseteq \mathbb{R}^d$, where, when restricted to a given $Q \in \mathcal{Q}$ we have

$$f|_Q(x) = V_Q x + A_Q \qquad (45)$$

where $V_Q \in \mathbb{R}^d, A_Q \in \mathbb{R}$ are the affine coefficients associated with the region $Q$. Then it is the case that $\nabla f : \mathbb{R}^d \to \mathbb{R}$ is a bounded function, where, aside from a set of measure 0, $M \subseteq \mathbb{R}^d$ (the boundaries of regions $Q$) where $\nabla f$ is not defined, one has

$$\|\nabla f(x)\|_2 \le \max_{Q \in \mathcal{Q}}\|V_Q\|_2, \ \ \forall x \in \mathbb{R}^d \backslash M. \qquad (46)$$

Thus, $\nabla f(x_0 + \delta)$ is a bounded random vector and therefore is sub-gaussian. $\qquad \square$

Given the operators $\lambda^+, \lambda^- : \mathbb{R}^{d \times d} \to \mathbb{R}$ as defined in the statement of Lemma 4 and fixed vector $\delta \in \mathbb{R}^d$, we denote $\lambda_\delta^+ := \lambda^+(A_\delta), \lambda_\delta^- := \lambda^-(A_\delta)$ and the corresponding unit eigenvectors as $v_\delta^+ \in \mathbb{R}^d$ and $v_\delta^- \in \mathbb{R}^d$ respectively. We denote random vectors $w_\delta^+, w_\delta^- \in \mathbb{R}^d$ by

$$w_\delta^+ = \sqrt{\lambda_\delta^+}\, v_\delta^+, \qquad\qquad w_\delta^- = \sqrt{-\lambda_\delta^-}\, v_\delta^- \qquad (47)$$

where $\sqrt{\lambda_\delta^+}$ and $v_\delta^+$ are a random variable random vector pair coming from the same $\delta$. An immediate consequence of Lemma 6 is that $w_\delta^+$ and $w_\delta^-$ are sub-gaussian random vectors:

**Lemma 7.** $w_\delta^+$ *and* $w_\delta^-$ *are sub-gaussian random-vectors*

*Proof.* Using Lemma 4(3), Lemma 4(2) and Lemma 6 we see that that $\lambda_\delta^+$ and $-\lambda_\delta^-$ are non-negative sub-exponential random variables and thus that $\sqrt{\lambda_\delta^+}$ and $\sqrt{-\lambda_\delta^-}$ are sub-gaussian random variables. We may say $w_\delta^+$ is a sub-gaussian random vector if $\langle w_\delta^+, z\rangle$ is a sub-gaussian random variable $\forall z \in \mathbb{R}^d$. Let us fix arbitrary $z \in \mathbb{R}^d$. As $\sqrt{\lambda_\delta^+}$ is sub-gaussian we have

$$\exists K_1 > 0 \ s.t. \ \mathbb{P}(|\sqrt{\lambda_\delta^+}| \ge t) \le 2\exp(-t^2/K_1^2) \,\forall t \ge 0 \qquad (48a)$$

Now, $\forall t \ge 0$

$$\mathbb{P}(|\langle w_\delta^+, z\rangle| \ge t) = \mathbb{P}(|\sqrt{\lambda_\delta^+}\langle v_\delta^+, z\rangle| \ge t) = \qquad (49a)$$

$$\mathbb{P}(|\sqrt{\lambda_\delta^+}| \ge t/|\langle v_\delta^+, z\rangle|) \overset{\text{C-S, }\|v_\delta^+\|_2=1}{\le} \mathbb{P}(|\sqrt{\lambda_\delta^+}| \ge t/\|z\|_2) \le 2\exp(-(t^2/\|z\|_2^2 K_1^2)) \qquad (49b)$$

Thus defining $K_1^{(z)} := K_1\|z\|_2$ we see that $\forall t \ge 0$

$$\mathbb{P}(|\langle w_\delta^+, z\rangle| \ge t) \le \exp(-t^2/(K_1^{(z)})^2). \qquad (50)$$

Thus $\langle w_\delta^+, z\rangle$ is sub-gaussian for arbitrary $z \in \mathbb{R}^d$. Therefore, $w_\delta^+$ is a sub-gaussian random-vector. The same argument holds to show that $w_\delta^-$ is a sub-gaussian random vector.

$\square$

Given any fixed $\delta$, it can be seen that

$$A_\delta = w_\delta^+ w_\delta^{+T} - w_\delta^- w_\delta^{-T}. \tag{51}$$

Thus one may re-write $H$ and $\hat{H}_n$ from Eq. 44 as:

$$H = \mathbb{E}_\delta[A_\delta] = \mathbb{E}_\delta[w_\delta^+ w_\delta^{+T} - w_\delta^- w_\delta^{-T}], \quad \hat{H}_n = \frac{1}{n}\sum_{i=1}^n A_{\delta_i} = \frac{1}{n}\sum_{i=1}^n w_{\delta_i}^+ w_{\delta_i}^{+T} - w_{\delta_i}^- w_{\delta_i}^{-T}. \tag{52}$$

Because $\{\delta_i\}_{i=1}^n$ are i.i.d. random vectors and $w_{\delta_i}^+, w_{\delta_i}^-$ are fully determined by $\delta_i$, it follows that $\{w_{\delta_i}^+\}_{i=1}^n$ and $\{w_{\delta_i}^-\}_{i=1}^n$ are both sets of i.i.d. random vectors.

We denote the following:

$$H^+ = \mathbb{E}_\delta[w_\delta^+ w_\delta^{+T}], \qquad\qquad H^- = \mathbb{E}_\delta[w_\delta^- w_\delta^{-T}], \tag{53a}$$

$$\hat{H}_n^+ = \frac{1}{n}\sum_{i=1}^n w_{\delta_i}^+ w_{\delta_i}^{+T}, \qquad\qquad \hat{H}_n^- = \frac{1}{n}\sum_{i=1}^n w_{\delta_i}^- w_{\delta_i}^{-T} \tag{53b}$$

It can be seen from the RHS of Eq. (52) that

$$\hat{H}_n = \hat{H}_n^+ - \hat{H}_n^-. \tag{54}$$

We now aim to decompose $H$ in terms of $H^+, H^-$, in order to derive separate concentration bounds. To this end, we prove the following lemma:

**Lemma 8.** $H^+$ and $H^-$ exist.

*Proof.* Let us consider the random matrix $w_\delta^+ w_\delta^{+T} \in \mathbb{R}^{d \times d}$. As $(w_\delta^+)_k \in \mathbb{R}$ is sub-gaussian $\forall k \in D$, the element $(w_\delta^+ w_\delta^{+T})_{ij} \in \mathbb{R}$ is sub-exponential as the product of two sub-gaussian's, $\forall i, j \in D$. Thus, $\mathbb{E}_\delta[(w_\delta^+ w_\delta^{+T})_{ij}]$ exists $\forall i, j \in D$. The same argument can be made to show $\mathbb{E}_\delta[(w_\delta^- w_\delta^{-T})_{ij}]$ exists $\forall i, j \in D$.

$\square$

In light of Lemma 8, the LHS of Eq. (52) may be decomposed as

$$H = \mathbb{E}_\delta[w_\delta^+ w_\delta^{+T} - w_\delta^- w_\delta^{-T}] = \mathbb{E}_\delta[w_\delta^+ w_\delta^{+T}] - \mathbb{E}_\delta[w_\delta^- w_\delta^{-T}] = H^+ - H^-. \tag{55}$$

Before deriving separate concentration bounds on $\hat{H}_n^+$ and $\hat{H}_n^-$ we note that

$$\lim_{n\to\infty} \hat{H}_n^+ = \mathbb{E}_\delta[w_\delta^+ w_\delta^{+T}], \qquad\qquad \lim_{n\to\infty} \hat{H}_n^- = \mathbb{E}_\delta[w_\delta^- w_\delta^{-T}]. \tag{56}$$

Finally, we bound the deviation of $\hat{H}_n^+$ and $\hat{H}_n^-$ from their expectations. Let us fix $\varepsilon, \gamma \in (0, 1]$. From Eq. (55), and the fact that $\{w_{\delta_i}^+\}_{i=1}^n$ is a set of i.i.d. sub-gaussian random vectors, Theorem 3.39, Remark 3.40 of Vershynin [87] may be applied, yielding: $\forall t \geq 0$

$$\mathbb{P}\left(\|\hat{H}_n^+ - H^+\|_2 > \max(\varepsilon_n^+, (\varepsilon_n^+)^2)\right) \leq 2\exp(-c^+ t^2) \tag{57}$$

where $\varepsilon_n^+ = C^+ \frac{\sqrt{d}}{\sqrt{n}} + \frac{t}{\sqrt{n}}$ and $C^+, c_+ > 0$ are constants depending on the sub-gaussian norm of $w_{\delta_i}^+$. Let us select $t = \sqrt{\frac{\log(4/\gamma)}{c^+}}$. Plugging into Eq. (57), we get

$$\mathbb{P}\left(\|\hat{H}_n^+ - H^+\|_2 > \max(C^+ \frac{\sqrt{d}}{\sqrt{n}} + \frac{\sqrt{\frac{\log(4/\gamma)}{c^+}}}{\sqrt{n}}, (C^+ \frac{\sqrt{d}}{\sqrt{n}} + \frac{\sqrt{\frac{\log(4/\gamma)}{c^+}}}{\sqrt{n}})^2)\right) \leq \frac{\gamma}{2} \tag{58}$$

Let us consider $n^+ = \frac{4}{\varepsilon^2}(C^+\sqrt{d} + \sqrt{\frac{\log(4/\gamma)}{c^+}})^2$. One may see that

$$\varepsilon_{n^+}^+ = C^+\frac{\sqrt{d}}{\sqrt{n^+}} + \frac{\sqrt{\frac{\log(4/\gamma)}{c^+}}}{\sqrt{n^+}} = \tag{59a}$$

$$C^+\frac{\varepsilon\sqrt{d}}{2(C^+\sqrt{d} + \sqrt{\frac{\log(4/\gamma)}{c^+}})} + \frac{\varepsilon\sqrt{\frac{\log(4/\gamma)}{c^+}}}{2(C^+\sqrt{d} + \sqrt{\frac{\log(4/\gamma)}{c^+}})} = \tag{59b}$$

$$\frac{\varepsilon}{2}\frac{C^+\sqrt{d} + \sqrt{\frac{\log(4/\gamma)}{c^+}}}{C^+\sqrt{d} + \sqrt{\frac{\log(4/\gamma)}{c^+}}} = \frac{\varepsilon}{2} \tag{59c}$$

Thus, given $n = n^+$ one has $\varepsilon_n^+ = \max(\frac{\varepsilon}{2}, (\frac{\varepsilon}{2})^2) = \frac{\varepsilon}{2}$, because $\frac{\varepsilon}{2} < \varepsilon \le 1$, and that

$$\mathbb{P}\left(\|\hat{H}_n^+ - H^+\|_2 > \frac{\varepsilon}{2}\right) \le \frac{\gamma}{2}. \tag{60}$$

In fact, because $\varepsilon_n^+$ is monotonically decreasing in $n$, given $n \ge n^+$ Eq. (60) holds.

The same logic above may be used to show that there exists constants $C^-, c^- > 0$ depending on the sub-gaussian norm of $w_\delta^-$ such that, given $n \ge n^- = \frac{4}{\varepsilon^2}(C^-\sqrt{d} + \sqrt{\frac{\log(4/\gamma)}{c^-}})^2$ one has

$$\mathbb{P}\left(\|\hat{H}_n^- - H^-\|_2 > \frac{\varepsilon}{2}\right) \le \frac{\gamma}{2}. \tag{61}$$

Finally, we combine the two bounds from Eq. (61) and Eq. (60). Given $n \ge \max(\frac{4}{\varepsilon^2}(C^+\sqrt{d} + \sqrt{\frac{\log(4/\gamma)}{c^+}})^2, \frac{4}{\varepsilon^2}(C^-\sqrt{d} + \sqrt{\frac{\log(4/\gamma)}{c^-}})^2)$ we have

$$\mathbb{P}(\|\hat{H}_n - H\|_2 > \varepsilon) = \mathbb{P}(\|\hat{H}_n^+ - \hat{H}_n^- - H^+ + H^-\|_2 > \varepsilon) = \quad \text{(Eq. (55)Eq. (54))} \tag{62a}$$

$$\mathbb{P}(\|(\hat{H}_n^+ - H^+) - (\hat{H}_n^- - H^-)\|_2 > \varepsilon) \le \tag{62b}$$

$$\mathbb{P}(\|(\hat{H}_n^+ - H^+)\|_2 + \|(\hat{H}_n^- - H^-)\|_2 > \varepsilon) \le \quad (\triangle - \text{Ineq.}) \tag{62c}$$

$$\mathbb{P}(\|\hat{H}_n^+ - H^+\|_2 > \frac{\varepsilon}{2} \cup \|\hat{H}_n^- - H^-\|_2 > \frac{\varepsilon}{2}) \le \tag{62d}$$

$$\mathbb{P}(\|\hat{H}_n^+ - H^+\|_2 > \frac{\varepsilon}{2}) + \mathbb{P}(\|\hat{H}_n^- - H^-\|_2 > \frac{\varepsilon}{2}) \le \quad \text{(Union bound)} \tag{62e}$$

$$\frac{\gamma}{2} + \frac{\gamma}{2} = \gamma \quad \text{(Eq. (60)Eq. (61))} \tag{62f}$$

$\square$

# D  Implementation Details

## D.1  Quadratic Optimization

Given a function $f : \mathbb{R}^d \to \mathbb{R}$, point $x_0 \in \mathbb{R}^d$ gradient Hessian pair $G \in \mathbb{R}^d, H \in \mathbb{R}^{d \times d}$ and a magnitude constraint $\varepsilon > 0$, we aim to solve the following optimization:

$$\min_{\Delta \in \mathbb{R}^d} f(x_0) + G^T(\Delta - x_0) + \frac{1}{2}(\Delta - x_0)^T H(\Delta - x_0), \quad s.t. \quad \|\Delta - x_0\|_2 \le \varepsilon, \tag{63}$$

as $f(x_0)$ is constant, the problem above is equivalent to

$$\min_{\delta \in \mathbb{R}^d} G^T\delta + \frac{1}{2}\delta^T H\delta, \quad s.t. \quad \|\delta\|_2 \le \varepsilon, \tag{64}$$

where we have replaced $\Delta - x_0$, which can be interpreted as the output after the attack with $\delta = \Delta - x_0$, the attack vector itself.

The optimization problem in Eq. (64) is non-convex as $H$ is not guaranteed to be positive semi-definite. However, as Slater's constraint qualification is satisfied, i.e. $\exists \delta \in \mathbb{R}^d \ s.t. \ \|\delta\|_2 < \varepsilon$, Eq. (64) may be solved exactly[12]. Specifically, the solution may be obtained by solving an equivalent convex optimization problem:

$$\min_{\gamma \in \mathbb{R}^d, X \in S^d} \mathbf{tr}(\frac{1}{2}HX) + G^T\gamma, \tag{65a}$$

$$s.t. \ \mathbf{tr}(X) - \varepsilon^2 \leq 0, \tag{65b}$$

$$[X, \gamma; \gamma^T, 1] \succeq 0 \tag{65c}$$

where $S^d$ denotes the the set of symmetric matrices and $\succeq$ indicates the block matrix $[X, \gamma; \gamma^T, 1] \in \mathbb{R}^{(d+1)\times(d+1)}$ is constrained to be positive semi-definite.

However, the optimization in Eq. (65) is expensive to solve when $d \gg 0$ as there are $\mathcal{O}(d^2)$ variables. For instance, MNIST and FMNIST have $d^2 \approx 6.0 \cdot 10^6$ and CIFAR10 has $d^2 \approx 10^8$. Thus, before converting Eq. (64) into Eq. (65), we elect to reduce the dimension of the optimization problem to $k \in \mathbb{N}, k \ll d$.

Let us consider the eigendecomposition $Q\Lambda Q^T = H$. Here the columns of $Q \in \mathbb{R}^{d \times d}$ are orthonormal eigenvectors of $H$. Given the $d$ eigenvalues $\{\lambda_1, \ldots, \lambda_d\}$, sorted such that $i < j \implies |\lambda_i| \geq |\lambda_j|$, we have $\Lambda = \mathrm{diag}(\lambda_1, \ldots, \lambda_d)$. We remove the last $d - k$ columns from $Q$ and $d - k$ columns and rows from $\Lambda$ to construct $\tilde{Q} \in \mathbb{R}^{d \times k}, \tilde{\Lambda} \in \mathbb{R}^{k \times k}$. Thus, we have a low-rank approximation of $H$:

$$H \simeq \tilde{Q}\tilde{\Lambda}\tilde{Q}^T. \tag{66}$$

Thus, Eq. (64) is approximately equivalent to another optimization which uses this low-rank approximation for $H$:

$$\min_{\delta \in \mathbb{R}^d} \ G^T\delta + \frac{1}{2}\delta^T\tilde{Q}\tilde{\Lambda}\tilde{Q}^T\delta, \quad s.t. \quad \|\delta\|_2 \leq \varepsilon. \tag{67}$$

Defining $\tilde{\delta} = \tilde{Q}^T\delta \in \mathbb{R}^k$, Eq. (67) is approximately equivalent to

$$\min_{\tilde{\delta} \in \mathbb{R}^k} \ G^T\tilde{Q}\tilde{\delta} + \frac{1}{2}\tilde{\delta}^T\tilde{\Lambda}\tilde{\delta}, \quad s.t. \quad \|\tilde{\delta}\|_2 \leq \varepsilon, \tag{68}$$

where the constraint is simplified to $\|\tilde{\delta}\|_2 \leq \varepsilon$ from $\|\tilde{Q}\tilde{\delta}\|_2 \leq \varepsilon$ as

$$\|\tilde{Q}\tilde{\delta}\|_2 = ((\tilde{Q}\tilde{\delta})^T(\tilde{Q}\tilde{\delta}))^{\frac{1}{2}} = (\tilde{\delta}^T\tilde{Q}^T\tilde{Q}\tilde{\delta})^{\frac{1}{2}} = (\tilde{\delta}^T\tilde{Q}^T\tilde{Q}\tilde{\delta})^{\frac{1}{2}} = (\tilde{\delta}^TI\tilde{\delta})^{\frac{1}{2}} = \|\tilde{\delta}\|_2. \tag{69}$$

Finally, $\tilde{\delta}^* \in \mathbb{R}^k$, the optimal solution to Eq. (68), is projected back to $\mathbb{R}^d$ yielding an approximate solution to Eq. (67):

$$\delta^* = \tilde{Q}\tilde{\delta}^* \in \mathbb{R}^d. \tag{70}$$

**Choosing k:** The choice of $k \in \mathbb{N}$ is determined using a threshold hyperparameter $T \in (0, 1]$. Given $T$, $k$ is chosen to be the smallest number of (sorted) eigenvalues/eigenvectors which account for a proportion of the total eigenvalue magnitude that is at least $T$:

$$k = \underset{k'}{\mathrm{argmin}}\{k' \in \mathbb{N} : \sum_{i=1}^{k'} |\lambda_i| \geq T \sum_{i=1}^{d} |\lambda_i|\}. \tag{71}$$

For the values of $T$ used for each dataset see App. E

### D.1.1 Similarities and Differences with CASO/CAFO

Our quadratic optimization is closely connected to the CAFO/CASO explanation vectors proposed by Singla et al. [74]. Both methods use optimizations to minimize a value outputted by the network, and the CASO method also uses a second-order Taylor expansion in their objective. Additionally, the proposed Smooth-CASO method is equivalent to an attack using SmoothGrad and SmoothHess on the cross-entropy loss function (when their regularizers are set to 0), which admits higher order derivatives.

However, there are some key differences. First, our optimization is meant for use with arbitrary network outputs as opposed to just the loss, as is the case with CAFO/CASO. While the vanilla loss Hessian modeled by Singla et al. [74] is proven to be positive semi-definite, this is not the case for arbitrary functions which one may wish to attack. Thus methods to optimize non-convex quadratic objectives, such as those outlined in App. D.1, are required. Second, our goal is not to generate an explanatory feature importance vector for analysis, as is an important motivation for CAFO/CASO, but to assess the quality of the gradient Hessian pair used to attack the function. For this reason, we do not use sparsity constraints such as in Singla et al. [74], which are in part meant to improve the interpretability of CAFO/CASO as explainers. Last we stress that the techniques used to find the cross-entropy loss Hessian for CASO and Smooth-CASO *cannot be used* for internal neurons, logits or regression valued output in ReLU networks, due to their piecewise-linearity.

### D.2 SmoothHess

The SmoothHess estimation procedure, and the amortization with SmoothGrad estimation, is presented below in Algorithm 1. While empirically our SmoothHess estimator converges, we have additionally found that reflecting each point $\delta_i$ in the perturbation set $\{\delta_i\}_{i=1}^n$ about the origin to create an augmented perturbation set $\{\delta_i\}_{i=1}^n \cup \{-\delta_i\}_{i=1}^n$ before estimation can result in faster per-sample convergence.

---

**Algorithm 1** Joint SmoothHess and SmoothGrad Estimation

---

**Input :** Sample of interest $x \in \mathbb{R}^d$, Neural network indexed to output scalar of interest $f : \mathbb{R}^d \to \mathbb{R}$, Covariance $\Sigma \in \mathbb{R}^{d \times d}$, Batch size for gradient oracle calls $n_1 \in \mathbb{N}$, Number of batches $n_2 \in \mathbb{N}$.
**Output :** $\hat{H}$ an estimate of SmoothHess, $\hat{G}$ an estimate of SmoothGrad

$\hat{H}, \hat{G} \leftarrow$ torch.zeros(d,d), torch.zeros(d) $\qquad \backslash\backslash \hat{H} \in \mathbb{R}^{d \times d}, \hat{G} \in \mathbb{R}^d$
$\Sigma^{-1} \leftarrow$ torch.inverse($\Sigma$) $\qquad \backslash\backslash \Sigma^{-1} \in \mathbb{R}^{d \times d}$, If $\Sigma$ diagonal $\mathcal{O}(d)$, Else $\mathcal{O}(d^3)$

$\backslash\backslash \mathcal{O}(n_1 n_2 (W + d^2))$ , $\mathcal{O}(W)$ is complexity of one forward pass through $f$
**for** $i = 1, \ldots, n_2$ **do**
$\quad \delta \leftarrow$ torch.normal($n_1, 0, \Sigma$) $\qquad \backslash\backslash \delta \in \mathbb{R}^{n_1 \times d}$
$\quad \nabla f(x + \delta) \leftarrow$ torch.autograd($f, x + \delta$) $\qquad \backslash\backslash \nabla f(x + \delta) \in \mathbb{R}^{n_1 \times d}, \mathcal{O}(n_1 W)$
$\quad \hat{A}_i \leftarrow$ torch.matmul($\delta, \nabla f(x + \delta).T$) $\qquad \backslash\backslash \hat{A}_i \in \mathbb{R}^{d \times d}, \mathcal{O}(n_1 d^2)$
$\quad \hat{H} \leftarrow \hat{H} + \hat{A}_i / n_1 n_2 \qquad \backslash\backslash \mathcal{O}(n_1 d^2)$
$\quad \hat{G} \leftarrow \hat{G} + \nabla f(x + \delta).\text{sum(dim = 0)} / n_1 n_2 \qquad \backslash\backslash \mathcal{O}(n_1 d)$
**end**
$\hat{H} \leftarrow$ torch.matmul($\Sigma^{-1}, \hat{H}$) $\qquad \backslash\backslash$ If $\Sigma$ diagonal $\mathcal{O}(d^2)$, Else $\mathcal{O}(d^3)$
$\hat{H} \leftarrow \hat{H} + \hat{H}.T \qquad \backslash\backslash \mathcal{O}(d^2)$

Return $\hat{H}, \hat{G}$

---

### D.3 Alternative Covariance Matrices

While in this work we use isotropic covariance matrices of the form $\Sigma = \sigma^2 I, \sigma > 0$, SmoothHess can be estimated using arbitrary positive definite covariance matrices $\Sigma \in \mathbb{R}^{d \times d}$. Such a covariance matrix can be set according to the users preference, encoded using the fact that the eigenvectors of $\Sigma$ represent directions of interest and their corresponding eigenvalues represent levels of smoothing.

In the simplest case the user already has an orthornomal eigenvector basis in mind, $a_1, \ldots, a_d \in \mathbb{R}^d$ as well as desired levels of smoothing $\sigma_1, \ldots, \sigma_d > 0$. In this case $\Sigma$ may be set by constructing eigenvector and eigenvalue matrices $Q = [a_1 | \ldots | a_d] \in \mathbb{R}^{d \times d}$ and $\Lambda = \text{diag}(\sigma_1, \ldots, \sigma_d) \in \mathbb{R}^{d \times d}$ and simply multiplying $\Sigma = Q \Lambda Q^T$.

Alternatively, the user may only have $k < d$ orthonormal eigenvectors in mind, $a_1, \ldots, a_k \in \mathbb{R}^d$, for which they wish to smooth at specific levels $\sigma_1, \ldots, \sigma_k > 0$. A procedure such as Gram-Schmidt may be used to find $a_{k+1}, \ldots, a_d \in \mathbb{R}^d$ which extends $a_1, \ldots, a_k$ to an orthonormal basis of $\mathbb{R}^d$. Again, the smoothing levels along the eigenvectors $a_{k+1}, \ldots, a_d$ may be chosen according to user preference. For instance, one may select $\sigma_{k+1} = \ldots = \sigma_d \approx 0$ if minimal smoothing is desired along these directions. Just as above $\Sigma$ may be set by constructing eigenvector and eigenvalue matrices $Q = [a_1 | \ldots | a_d] \in \mathbb{R}^{d \times d}$ and $\Lambda = \text{diag}(\sigma_1, \ldots, \sigma_d) \in \mathbb{R}^{d \times d}$ and simply multiplying $\Sigma = Q \Lambda Q^T$.

# E  Experiment Setup

## E.1  Datasets and Models

In this work we make use of six datasets, two synthetic datasets (Four Quadrants, Nested Interactions) three benchmark datasets (MNIST, FMNIST, CIFAR10) and a real world medical dataset (Spirometry). Below we describe these datasets and training details.

**Four Quadrant**. The Four Quadrant dataset consists of points $x \in \mathbb{R}^2$ sampled from the grid $[-2, 2] \times [-2, 2]$ with a spacing of $0.008$. A 6-layer fully connected ReLU network was trained using RMSProp [83] on the Four Quadrant dataset achieving a final mean-squared-error of $\approx 1e\text{-}4$. Training lasted for $40,000$ iterations with a batch size of $128$ and a starting learning rate of $1e\text{-}3$ which was decayed by a factor of $1e\text{-}1$ at iterations $5000$, $10,000$ and $20,000$.

**Nested Interactions**. The Nested Interactions dataset consists of points $x \in \mathbb{R}^2$ sampled from the grid $[-2, 2] \times [-2, 2]$ with a spacing of $0.008$. A 6-layer fully connected ReLU network was trained using RMSProp on the Nested Interactions dataset achieving a final mean-squared-error of $\approx 1e\text{-}1$. Training lasted for $200,000$ iterations with a batch size of $64$ and a starting learning rate of $1e\text{-}3$ which was decayed by a factor of $1e\text{-}1$ at iterations $40,000, 80,000, 120,000$ and $160,000$. For more details on Nested Interactions see App. F.3.

**MNIST** MNIST consists of 70,000 28x28 greyscale images, each corresponding with one of the digits 0-9. There are $60,000$ and $10,000$ images in the pre-defined train and test sets respectively. We further split the train set into $50,000$ images for training and $10,000$ for validation. A 5-layer fully connected network with dimensions 500-300-250-250-250 was trained with stochastic gradient descent for 30 epochs, with a batch size of $128$ and a starting learning rate of $1e\text{-}2$ which was decayed by a factor of $1e\text{-}1$ at iterations $4,000$ and $8,000$. Final accuracies of $\approx 100\%$ (Train) $\approx 100\%$ (Val) and $\approx 98\%$ (Test) were achieved. All images are flattened. For $\mathcal{P}_{MSE}$ all methods were evaluated on 200 test points. For adversarial attacks all methods were evaluated on 400 test points. The threshold for the quadratic optimization attack was set to $T = 0.98$.

**FMNIST**. FMNIST consists of 70,000 28x28 greyscale images, each corresponding with one of 10 articles of clothing. There are $60,000$ and $10,000$ images in the pre-defined train and test sets respectively. We further split the train set into $50,000$ images for training and $10,000$ for validation. Final accuracies of $\approx 93\%$ (Train) $\approx 93\%$ (Val) and $\approx 88\%$ (Test) were achieved. The network and training details are identical to that used for MNIST above. All images are flattened. For $\mathcal{P}_{MSE}$ all methods were evaluated on 200 test points. For adversarial attacks all methods were evaluated on 400 test points. The threshold for the quadratic optimization attack was set to $T = 0.98$.

**CIFAR10** CIFAR10 consists of 60,000 3x32x32 RGB color images, each corresponding with an animal or vehicle. There are 50,000 and 10,000 images in the pre-defined train and test sets respectively. We further split the train set into $40,000$ images for training and $10,000$ for validation. A ResNet-18 [32] was trained on CIFAR10 for 55 epochs using a batch size of $128$. The first 5 epochs were used for warmup with a starting learning rate of $1e\text{-}2$ ending at $0.5$. For the rest of training a cosine decay schedule was used [52] decaying down to $1e\text{-}5$ by the final epoch. Augmentations used for training were (i) Random Horizontal Flip ($p = 0.5$) (ii) Color Jitter ($p = 0.8$) with brightness, contrast and saturation values equal to $0.4$ and hue value $0.1$ (iii) Random Grayscale ($p = 0.2$). Final accuracies of $\approx 85\%$ (Train, Augmentations) $\approx 96\%$ (Val, No Augmentations) and $\approx 90\%$ (Test, No Augmentations) were achieved. No augmentations were applied to the validation/test data when evaluating explainers. For $\mathcal{P}_{MSE}$ all methods were evaluated on 100 test points. For adversarial attacks all methods were evaluated on 200 test points. The threshold for the quadratic optimization attack was set to $T = 0.8$

**Spirometry.** The Spirometry dataset uses raw exhalation curves, measured in volume over time, recorded during a spirometry exam. Each spirometry curve is measured in 10ms intervals, which we downsample to 50ms intervals and limit to 15s in total length, resulting in 300 features. We use the UK Biobank dataset, which is a large, population-based study conducted in the United Kingdom. Participant statistics have been previously reported in Sudlow et al. [79]. The UK Biobank records 2-3 exhalation efforts for each participant, using a Vitalograph Pneumotrac 6800 device*. If two efforts are recorded as passing acceptability criteria and are also reproducible ($\leq 5\%$ difference in Forced Vital Capacity (FVC) and Forced Expiratory Volume in 1 Second (FEV$_1$)), then the third

---

*https://biobank.ctsu.ox.ac.uk/crystal/crystal/docs/Spirometry.pdf

| Dataset | MNIST | | | | | | FMNIST | | | | | | CIFAR10 | | | | | |
|---|---|---|---|---|---|---|---|---|---|---|---|---|---|---|---|---|---|---|
| *Function* | Class Logit (↓) | | | Int. Neuron (↓) | | | Class Logit (↓) | | | Int. Neuron (↓) | | | Class Logit (↓) | | | Int. Neuron (↓) | | |
| $\epsilon$ | 0.25 | 0.50 | 1.00 | 0.25 | 0.50 | 1.00 | 0.25 | 0.50 | 1.00 | 0.25 | 0.50 | 1.00 | 0.25 | 0.50 | 1.00 | 0.25 | 0.50 | 1.00 |
| SH+SG (Us) | 4.5e-5 | 1.8e-4 | 7.2e-4 | 4.5e-5 | 1.8e-4 | 7.2e-4 | 4.5e-5 | 1.8e-4 | 7.2e-4 | 4.5e-5 | 1.8e-4 | 7.2e-4 | 1.1e-5 | 4.6e-5 | 1.8e-4 | 1.1e-5 | 4.6e-5 | 1.8e-4 |
| SG [76] | 4.5e-5 | 1.8e-4 | 7.2e-4 | 4.5e-5 | 1.8e-4 | 7.2e-4 | 4.5e-5 | 1.8e-4 | 7.2e-4 | 4.5e-5 | 1.8e-4 | 7.2e-4 | 1.1e-5 | 4.6e-5 | 1.8e-4 | 1.1e-5 | 4.6e-5 | 1.8e-4 |
| SP H + G | 400.0 | 200.0 | 95.0 | 400.0 | 200.0 | 100.0 | 350.0 | 160.0 | 75.0 | 360.0 | 190.0 | 95.0 | 11.5 | 6.0 | 3.5 | 11.0 | 6.0 | 4.0 |
| SP G | 390.0 | 200.0 | 100.0 | 400.0 | 200.0 | 100.0 | 390.0 | 180.0 | 95.0 | 360.0 | 190.0 | 95.0 | 14.0 | 8.0 | 4.0 | 16.0 | 7.0 | 4.0 |
| SW (H + G) | 190.0 | 95.0 | 55.0 | 190.0 | 95.0 | 50.0 | 170.0 | 80.0 | 45.0 | 170.0 | 90.0 | 45.0 | 8.5 | 8.5 | 8.5 | 11.0 | 11.0 | 11.0 |
| SW G | 190.0 | 95.0 | 55.0 | 190.0 | 95.0 | 50.0 | 180.0 | 100.0 | 65.0 | 170.0 | 90.0 | 50.0 | 8.5 | 8.5 | 8.5 | 11.0 | 11.0 | 11.0 |

Table 3: Selected values of $\sigma^2$ and $\beta$, achieving the lowest average $\mathcal{P}_{MSE}$ on on a held-out validation set.

effort is omitted. A common metric used to evaluate lung health is the Forced Expiratory Volume in 1 Second ($FEV_1$), which is the maximum volume of air that can be expelled by the participant in 1 second [59]. Note that a participant's $FEV_1$ measurement is taken as the maximum $FEV_1$ over all recorded exhalation efforts during a single visit.

The spirometer automatically evaluates effort against a number of acceptability criteria. One such criteria is the detection of coughing. In our experiment, we use a subset of exhalation efforts where coughing was detected and train a CNN to predict the participant's final $FEV_1$ measurement. This subset contains 8,721 samples, which we split into training (80%) and test (20%) partitions. We follow the preprocessing in Hill et al. [36] to ensure that participants have at least one effort that passes quality control where $FEV_1$ can be measured. The trained CNN includes 10 convolution blocks. Each block contains a 1-d convolution of kernel width 200 and 20 channels, batch normalization, dropout (p=0.5), and skip connection. The model is trained using mean squared error (MSE), achieving 0.563 MSE on the train set and 0.547 MSE on the test set.

## E.2    Hyperparameters

For ease of reading, we use the following notation below: $[a : b : c] = \{a, a + c, a + 2c, \ldots, b - 2c, b - c\} \subseteq \mathbb{R}$ denotes the set of points between $a$ (inclusive) and $b$ (exclusive) at intervals of size $c$. Here $a$ and $b$ are chosen such that $a < b$ and $(b - a) \mod c = 0$. An example is: $[0.1 : 1.0 : 0.1] = \{0.1, 0.2, 0.3, 0.4, 0.5, 0.6, 0.7, 0.8, 0.9\}$.

### E.2.1    $\mathcal{P}_{MSE}$

For each of the 18 combinations of dataset, function and neighborhood size $\varepsilon$ the performance of $\beta$ and $\sigma$ are validated on a held out set before selection. 200 validation points are used to choose $\sigma, \beta$ for MNIST and FMNIST and 50 validation points are used for CIFAR10. For SmoothHess + SmoothGrad and SmoothGrad *only three values of $\sigma$* are validated, based on the common sense criterion $\sigma = \varepsilon/\sqrt{d}$: given neighborhood size $\varepsilon$ and dataset with dimension $d$, $\sigma$ is chosen from $\sigma \in \{\varepsilon/2\sqrt{d}, 3\varepsilon/4\sqrt{d}, \varepsilon/\sqrt{d}\}$.

The following values of $\beta$ are checked on a validation set:

**MNIST and FMNIST** $\beta \in [0.1 : 1 : 0.1] \cup [1 : 20 : 1] \cup [20 : 95 : 5] \cup [100 : 800 : 10]$

**CIFAR10:** $\beta \in [0.1 : 1 : 0.1] \cup [1 : 20 : 0.5] \cup [20 : 95 : 5] \cup [100 : 800 : 10]$

The values of $\sigma$ for SH and SH + SG, and $\beta$ for SP H + G and SP G which achieve the lowest $\mathcal{P}_{MSE}$ on the validation data are shown in Table 3. The results for SoftPlus $\beta$ are interesting: (i) We see that MNIST and FMNIST results, for both class logit and interior neuron, are consistent for fixed $\varepsilon$. Further we see that CIFAR10 results between class logit and interior neuron are consistent for fixed $\varepsilon$. (ii) The optimal value of $\beta$ seems to be approximately proportional to the value of $\varepsilon$. While our results in Table 1 show that SmoothHess is better at capturing local interactions than the SoftPlus Hessian, the results in Table 3 indicate that the relationship between $f$, $f_\beta$ and $\beta$ warrants further exploration.

### E.2.2    Adversarial Attacks

200 validation points were used to choose $\beta$ and $\sigma^2$ for MNIST and FMNIST and 50 validation points were used to choose $\beta$ and $\sigma^2$ for CIFAR10.

| Dataset | MNIST | | | | | FMNIST | | | | | CIFAR10 | | | | |
|---|---|---|---|---|---|---|---|---|---|---|---|---|---|---|---|
| *Attack Magnitude* $\epsilon$ | 0.25 | 0.50 | 0.75 | 1.25 | 1.75 | 0.25 | 0.50 | 0.75 | 1.25 | 1.75 | 0.1 | 0.2 | 0.3 | 0.4 | 1.0 |
| SH+SG (Us) | 1e-3 | 5e-3 | 3e-2 | 5e-2 | 1.5e-1 | 1e-3 | 4e-2 | 3e-2 | 8e-2 | 3e-2 | 5e-5 | 1e-4 | 4e-4 | 4e-4 | 7.5e-4 |
| SG [76] | 1e-3 | 5e-3 | 2e-2 | 6e-2 | 8e-2 | 5e-3 | 4e-2 | 8e-2 | 1.5e-1 | 3e-1 | 5e-5 | 3e-4 | 5e-4 | 7.5e-4 | 3e-3 |
| SP H + G | 8 | 18 | 17 | 10 | 6 | 20 | 25 | 13 | 9 | 6 | 5 | 4 | 3 | 2 | 2 |
| SP G | 10 | 12 | 10 | 11 | 3 | 30 | 9 | 11 | 8 | 3 | 5 | 4 | 3 | 2 | 2 |

Table 4: Selected values of $\sigma^2$ and $\beta$, achieving the lowest post-hoc accuracy of adversarial attacks on a held-out validation set.

The following values of $\sigma^2$ were validated for adversarial attacks:

**MNIST and FMNIST:** $\sigma^2 \in \{0.001, 0.005\} \cup [0.01 : 0.1 : 0.01] \cup [0.15 : 1.00 : 0.05]$

**CIFAR10:** $\sigma^2 \in \{5e\text{-}05, 7.5e\text{-}05\} \cup [0.0001 : 0.0006 : 0.0001] \cup \{0.00075\} \cup [0.001 : 0.006 : 0.001]$

The following values of $\beta$ were validated for adversarial attacks:

**MNIST and FMNIST:** $\beta \in [1 : 20 : 1] \cup [20 : 100 : 5] \cup [100 : 210 : 10]$.

**CIFAR10:** $\beta \in [1 : 10 : 1] \cup [10 : 45 : 5]$

The values of $\sigma^2$ and $\beta$ that achieve lowest validation post-hoc accuracy are shown in Table 4. Following intuition, it is generally the case that parameters corresponding with increased smoothing (larger $\sigma^2$ and smaller $\beta$) achieve better results (lower post-hoc accuracy) for large $\varepsilon$, and parameters corresponding to less smoothing (smaller $\sigma^2$ and larger $\beta$) achieve better results for small $\varepsilon$.

# F  Additional Experiments

## F.1  $\mathcal{P}_{\mathcal{MSE}}$

**Comparison with Swish:** In Table 1, $\mathcal{P}_{MSE}$ results are presented for five methods: the first (SmoothGrad) and second (SmoothHess + SmoothGrad) order Taylor expansions of the ReLU network $f$ convolved with a Gaussian, the first and second order Taylor expansions of the SoftPlus smoothed network and the vanilla (unsmoothed) Gradient. We present Table 5, a version of Table 1 which includes additional results comparing with Swish [65] smoothed networks. Swish, an alternative smooth activation to SoftPlus, is formally defined as $\mathrm{Sw}_\beta(x) = x \, \mathrm{sigmoid}(\beta x)$ where $\mathrm{sigmoid}(x) = \frac{1}{1+\exp{(-x)}}$ and $\beta$ is a hyperparamter determining the level of smoothing. It can be seen in Table 5 that Swish is generally less effective then SoftPlus, and is outperformed by our method at each combination of dataset and locality.

**Standard Deviation:** We report the standard deviation of the $\mathcal{P}_{MSE}$ for each method, dataset, function and neighborhood size $\varepsilon$ in Table 6. Of the 18 $\mathcal{P}_{MSE}$ results, SmoothHess + SmoothGrad attains the lowest standard deviation for 15 and ties with SoftPlus Hessian + SoftPlus Gradient for 2. The standard deviation of SoftPlus Hessian + SoftPlus Gradient is the lowest for FMNIST internal neuron at $\varepsilon = 0.5$, achieving 2.4e-7 while SmoothHess + SmoothGrad achieves 2.5e-7.

**ResNet101:** We repeat our $\mathcal{P}_{MSE}$ experiment for the predicted class logits of CIFAR10 using a ResNet101, reporting results in Table 7. It can be seen in the leftmost column that our method, SH + SG, achieves superior performance to the competing methods at each locality, indicating that SmoothHess can generalize to larger network architectures. The standard deviation of $P_{MSE}$, as well as choices of $\sigma^2$ / $\beta$ (as selected from a validation set) are reported in the center and rigthmost columns, respectively.

## F.2  Adversarial Attacks

**Comparison with Vanilla Hessian:** One may use the vanilla Hessian of the predicted SoftMax probability, which admits higher order derivatives, to construct adversarial attacks. Table 8 is an updated version of Table 2 which includes results for attacks using the vanilla Hessian + vanilla Gradient (H + G), in the third to last row.

We see that inclusion of the vanilla Hessian generally results in more effective attacks then use of the vanilla gradient alone. H + G ties SH+SG and SP H + G for lowest post-hoc accuracy at the

| Dataset | MNIST | | | | | | FMNIST | | | | | | CIFAR10 | | | | | |
|---|---|---|---|---|---|---|---|---|---|---|---|---|---|---|---|---|---|---|
| *Function* | Class Logit ($\downarrow$) | | | Int. Neuron ($\downarrow$) | | | Class Logit ($\downarrow$) | | | Int. Neuron ($\downarrow$) | | | Class Logit ($\downarrow$) | | | Int. Neuron ($\downarrow$) | | |
| $\epsilon$ | 0.25 | 0.50 | 1.00 | 0.25 | 0.50 | 1.00 | 0.25 | 0.50 | 1.00 | 0.25 | 0.50 | 1.00 | 0.25 | 0.50 | 1.00 | 0.25 | 0.50 | 1.00 |
| SH+SG (Us) | **9.6e-7** | **7.8-6** | **6.7e-5** | **4.9e-8** | **4.0e-7** | **3.3e-6** | **6.5e-7** | **4.0e-6** | **4.3e-5** | **2.0e-8** | **1.8e-7** | **1.6e-6** | **9.8e-4** | **2.2e-2** | **1.2e-1** | **8.1e-4** | **1.4e-5** | **1.6e-4** |
| SG [76] | 4.5e-6 | 4.1e-5 | 3.9e-4 | 2.1e-7 | 1.7e-6 | 1.5e-5 | 3.0e-6 | 2.7e-5 | 2.6e-4 | 1.0e-7 | 9.0e-7 | 7.0e-6 | 1.3e-2 | 8.6e-2 | 4.9e-1 | 1.3e-5 | 1.1e-4 | 8.3e-4 |
| SP (H + G) | 1.2e-6 | 9.6e-6 | 8.1e-5 | 5.5e-8 | 4.4e-7 | 3.7e-6 | 9.6e-7 | 7.5e-6 | 6.5e-5 | 3.0e-8 | 2.1e-7 | 1.8e-6 | 2.1e-3 | 3.3e-2 | 2.5e-1 | 1.1e-5 | 1.0e-4 | 7.0e-4 |
| SP G | 4.6e-6 | 4.1e-5 | 3.9e-4 | 2.1e-7 | 1.7e-6 | 1.5e-5 | 3.2e-6 | 2.8e-5 | 2.6e-4 | 1.0e-7 | 8.5e-7 | 7.2e-6 | 1.3e-2 | 9.0e-2 | 5.2e-1 | 5.1e-5 | 2.9e-4 | 1.6e-3 |
| SW (H+ G) | 2.4e-6 | 2.0e-5 | 1.9e-4 | 1.0e-7 | 8.3e-7 | 7.3e-6 | 2.1e-6 | 1.7e-5 | 1.8e-4 | 5.0e-8 | 4.3e-7 | 3.7e-6 | 1.1e-2 | 3.3e-1 | 7.9e0 | 6.2e-5 | 1.7e-3 | 3.8e-2 |
| SW G | 5.6e-6 | 5.0e-5 | 4.9e-4 | 2.4e-7 | 2.0e-6 | 1.8e-5 | 3.9e-6 | 3.5e-5 | 3.5e-4 | 1.1e-7 | 9.6e-7 | 8.3e-6 | 4.9e-2 | 9.8e-2 | 6.0e-1 | 5.6e-5 | 3.4e-4 | 2.0e-3 |
| G [73] | 4.2e-3 | 1.7e-2 | 6.7e-2 | 2.0e-3 | 7.0e-3 | 2.9e-2 | 3.8e-3 | 1.5e-2 | 6.0e-2 | 1.0e-4 | 4.0e-4 | 1.8e-3 | 3.0e-1 | 1.2e-0 | 5.0e-0 | 9.0e-4 | 3.5e-3 | 1.4e-2 |

Table 5: Average $\mathcal{P}_{MSE}$ results at three radii $\varepsilon$, with the inclusion of vanilla Gradient (G). Other methods include: SmoothHess + SmoothGrad (SH+SG,Us) SmoothGrad (SG) SoftPlus Grad (SP G) SoftPlus Hessian + Gradient (SP H + G). Results are provided for the predicted class logit, and the penultimate neuron maximally activated by the "three," dress and cat classes for MNIST, FMNIST and CIFAR10 respectively.

| Dataset | MNIST | | | | | | FMNIST | | | | | | CIFAR10 | | | | | |
|---|---|---|---|---|---|---|---|---|---|---|---|---|---|---|---|---|---|---|
| *Function* | Class Logit ($\downarrow$) | | | Int. Neuron ($\downarrow$) | | | Class Logit ($\downarrow$) | | | Int. Neuron ($\downarrow$) | | | Class Logit ($\downarrow$) | | | Int. Neuron ($\downarrow$) | | |
| $\epsilon$ | 0.25 | 0.50 | 1.00 | 0.25 | 0.50 | 1.00 | 0.25 | 0.50 | 1.00 | 0.25 | 0.50 | 1.00 | 0.25 | 0.50 | 1.00 | 0.25 | 0.50 | 1.00 |
| SH+SG (Us) | **9.9e-7** | **7.3-6** | **6.4e-5** | **3.9e-8** | **2.7e-7** | **2.1e-6** | **1.0e-6** | **7.7e-6** | **6.4e-5** | **2.8e-8** | 2.5e-7 | **1.9e-6** | **4.7e-3** | **1.1e-1** | **4.3e-1** | **1.5e-5** | **3.6e-5** | **4.2e-4** |
| SG [76] | 7.0e-6 | 7.4e-5 | 7.1e-5 | 2.2e-7 | 1.6e-6 | 1.2e-5 | 5.5e-6 | 5.1e-5 | 4.6e-4 | 2.0e-7 | 1.8e-6 | 1.3e-5 | 4.7e-2 | 1.9e-1 | 7.5e-1 | 4.2e-5 | 4.1e-4 | 2.9e-3 |
| SP (H + G) | 1.2e-6 | 9.5e-6 | 7.7e-5 | 4.4e-8 | 2.9e-7 | **2.1e-6** | 2.4e-6 | 1.5e-5 | 1.2e-4 | 3.1e-8 | **2.4e-7** | **1.9e-6** | 6.5e-3 | 1.3e-1 | 6.4e-1 | 5.7e-4 | 4.0e-4 | 1.9e-3 |
| SP G | 7.1e-6 | 7.4e-5 | 7.2e-4 | 2.2e-7 | 1.6e-6 | 1.2e-5 | 6.1e-6 | 5.4e-5 | 4.7e-4 | 2.0e-7 | 1.8e-6 | 1.4e-5 | 4.8e-2 | 1.9e-1 | 7.6e-1 | 2.6e-4 | 8.9e-4 | 3.6e-3 |
| SW (H+ G) | 2.6e-6 | 2.0e-5 | 1.9e-4 | 6.5e-8 | 4.6e-7 | 4.2e-6 | 7.2e-6 | 3.3e-5 | 3.7e-4 | 5.3e-8 | 4.5e-7 | 4.0e-6 | 5.0e-2 | 1.7e0 | 4.3e1 | 3.6e-4 | 1.2e-2 | 2.7e-1 |
| SW G | 7.3e-6 | 7.2e-5 | 7.4e-4 | 2.1e-7 | 1.5e-6 | 1.2e-5 | 6.6e-6 | 6.2e-5 | 6.2e-4 | 1.5e-7 | 1.1e-6 | 9.0e-6 | 5.0e-2 | 2.1e-1 | 8.6e-1 | 2.8e-4 | 1.0e-3 | 4.2e-3 |
| G [73] | 2.1e-3 | 8.4e-3 | 3.4e-2 | 7.9e-5 | 3.0e-4 | 1.2e-3 | 2.7e-3 | 1.1e-2 | 4.3e-2 | 9.1e-5 | 4.0e-4 | 1.4e-3 | 2.4e-1 | 9.6e-1 | 3.7e-0 | 9.0e-4 | 3.7e-3 | 1.5e-2 |

Table 6: Standard deviation of $\mathcal{P}_{MSE}$ at three radii $\epsilon$. SmoothHess + SmoothGrad (SH+SG,Us) SmoothGrad (SG) SoftPlus Grad (SP G) SoftPlus Hessian + Gradient (SP H + G) Vanilla gradient (G). Results are provided for the predicted class logit, and the penultimate neuron maximally activated by the "three," dress and cat classes for MNIST, FMNIST and CIFAR10 respectively.

smallest magnitude ($\varepsilon = 0.25$) for the simplest dataset (MNIST). However, as no smoothing is done, the attacks generated from the H + G are generally significantly weaker then those generated using smooth surrogates. In fact, aside from MNIST with $\varepsilon = 0.25$, the second order vanilla H + G attacks achieve higher post-hoc accuracy then first-order method SmoothGrad for all datasets and values of $\varepsilon$. This is especially apparent for CIFAR10, the most complex dataset.

### F.3 Nested Interactions

We use the Nested Interactions dataset to highlight SmoothHess's ability to capture different interactions occurring at *various localities* around a given point. In this experiment we measure interactions around the origin $x_0 = (0,0)^T \in \mathbb{R}^2$.

Just like the Four Quadrant dataset, the Nested Interactions dataset consists of points $x \in \mathbb{R}^2$ sampled uniformly from $[-2, 2] \times [-2, 2] \subset \mathbb{R}^2$ with a spacing of 0.008. We establish different interactions occurring around the origin $x_0$, based upon the distance from $x_0$. Specifically, we set the label for a given point $x$ by: $x \in B_{0.6}(x_0) \implies y(x) = \frac{1}{2}x_1^2 + x_1 x_2$ , $x \in B_{1.2}(x_0) \backslash B_{0.6}(x_0) \implies y(x) = x_1 x_2$, $x \in \mathbb{R}^2 \backslash B_{1.2}(x_0) \implies y(x) = -5x_1 x_2$.

In words, the interaction between features $x_1$ and $x_2$ is 1 inside the radius-1.2 ball around $x_0$ and is $-5$ outside of this ball. The interaction between $x_1$ and itself is 1 inside the radius-0.6 ball around $x_0$ and 0 outside of this ball. The interaction between $x_2$ and itself is 0 over all of $\mathbb{R}^2$.

We train a 6-layer neural network on the Nested Interactions dataset and estimate SmoothHess and SoftPlus Hessian for $\sigma^2 \in \{1e\text{-}6, \dots, 1e1\}$ and $\beta \in \{1e\text{-}1, \dots, 4.0\}$ respectively. The interaction results for $x_1$ with itself, $x_1$ with $x_2$ and $x_2$ with itself as a function of the level of smoothing ($\sigma$ or $\beta$) are reported in Figure 5.

As the target function $y(x)$ is discontinuous, it is not possible for a network to memorize the Nested Interactions dataset. Thus, there very well may be interactions occurring in the network which are not described as above; the interactions we know occur in the data are not a pure "gold-standard". That being said, Figure 5 shows that SmoothHess captures the interactions as we know occur in the data, and the SoftPlus Hessian does not. This suggests that, to a large extent, both (i) the network has

| Value | $\mathcal{P}_{\mathcal{MSE}}$ | | | $\mathcal{P}_{\mathcal{MSE}}$ Std | | | $\sigma^2, \beta$ used | | |
|---|---|---|---|---|---|---|---|---|---|
| $\epsilon$ | 0.25 | 0.50 | 1.00 | 0.25 | 0.50 | 1.00 | 0.25 | 0.50 | 1.00 |
| SH+SG (Us) | **1.3e-4** | **9.5e-4** | **6.9e-3** | **1.7e-4** | **1.1e-3** | **1.1e-2** | 1.1e-5 | 4.6e-6 | 1.8e-4 |
| SG | 5.3e-4 | 4.3e-3 | 3.3e-2 | 1.0e-3 | 8.3e-3 | 6.7e-2 | 1.1e-5 | 4.6e-6 | 1.8e-4 |
| SP (H + G) | 3.7e-4 | 2.6e-3 | 1.7e-2 | 6.6e-4 | 2.7e-3 | 3.1e-2 | 41.0 | 16.0 | 11.5 |
| SP G | 6.2e-4 | 5.0e-3 | 3.7e-2 | 1.1e-3 | 8.8e-3 | 6.9e-2 | 55.0 | 22.5 | 11.5 |
| Swish (H + G) | 1.5e-3 | 1.2e-2 | 4.5e-2 | 2.5e-3 | 8.3e-3 | 2.8e-2 | 27.5 | 17.5 | 1.5 |
| Swish G | 8.8e-4 | 7.0e-3 | 5.3e-2 | 1.4e-3 | 1.1e-2 | 8.3e-2 | 60.0 | 1.0e6 | 1.0e6 |
| G | 3.8e0 | 1.5e+1 | 6.0e+1 | 3.3e0 | 1.3e1 | 5.2e1 | n/a | n/a | n/a |

Table 7: Using a ResNet101 trained on CIFAR10, the average $\mathcal{P}_{MSE}$ achieved by SmoothHess + SmoothGrad (SH + SG, Us), SmoothGrad (SG), SoftPlus Hessian + SoftPlus Gradient (SP (H+G)), SoftPlus Gradient (SP G), Swish Hessian + Swish Gradient (Swish (H+G)), Swish Gradient (Swish G) and Vanilla Gradient (G), is evaluated as a proxy for explainer quality. Results are reported at three radii $\varepsilon$, for the predicted class logit. **Left:** $\mathcal{P}_{MSE}$ results are reported. The lowest value in each column is bolded. **Middle:** The standard deviation of $\mathcal{P}_{MSE}$ is reported. The lowest value in each column is bolded. **Right:** The smoothing hyperparameter ($\sigma^2$ for SmoothGrad and SmoothHess + SmoothGrad, $\beta$ for SoftPlus and Swish) used is reported. *Our method, SH + SG, achieves the lowest $\mathcal{P}_{MSE}$ for each of the three radii $\varepsilon$.* This indicates that the interactions SmoothHess captures improve the model of network behaviour, *even for large networks such as ResNet101.*

| *Dataset* | **MNIST** | | | | | **FMNIST** | | | | | **CIFAR10** | | | | |
|---|---|---|---|---|---|---|---|---|---|---|---|---|---|---|---|
| *Attack Magnitude $\epsilon$* | 0.25 | 0.50 | 0.75 | 1.25 | 1.75 | 0.25 | 0.50 | 0.75 | 1.25 | 1.75 | 0.1 | 0.2 | 0.3 | 0.4 | 1.0 |
| SH+SG (Us) | **93.0** | 80.3 | 48.0 | 10.5 | 2.0 | **79.5** | 46.8 | 25.0 | 3.5 | **0.0** | 62.5 | 38.5 | 26.5 | 15.0 | 4.5 |
| SG [76] | 93.3 | 81.8 | 48.8 | 11.3 | 2.8 | **79.5** | 49.3 | 26.3 | 4.0 | **0.0** | 65.0 | 42.0 | 27.5 | 17.0 | **0.0** |
| SP (H + G) | **93.0** | 81.8 | 51.5 | 15.8 | 7.5 | 79.8 | 51.0 | 27.5 | 5.3 | 0.8 | 64.5 | 42.0 | 31.0 | 23.5 | 7.5 |
| SP G | 93.3 | 82.3 | 53.8 | 16.3 | 5.0 | 79.8 | 51.5 | 29.5 | 7.8 | 1.0 | 66.5 | 47.5 | 36.0 | 29.5 | 8.5 |
| H + G | **93.0** | 81.8 | 55.3 | 19.0 | 11.8 | 80.0 | 50.0 | 30.3 | 9.5 | 2.0 | 68.0 | 51.5 | 40.5 | 32.5 | 22.0 |
| G [73] | 93.3 | 82.8 | 56.0 | 18.5 | 8.8 | 80.3 | 52.3 | 31.8 | 11.0 | 2.5 | 69.0 | 51.5 | 41.0 | 34.0 | 21.5 |
| Random | 99.8 | 99.5 | 99.0 | 99.0 | 98.8 | 99.3 | 98.0 | 97.3 | 95.5 | 93.8 | 100.0 | 99.5 | 99.0 | 98.5 | 96.5 |

Table 8: Post-hoc accuracy of adversarial attacks performed on the predicted SoftMax probability, at five attack magnitudes $\epsilon$, with the inclusion of the vanilla Hessian + vanilla Gradient (H + G). Lower is better. Other methods include: SmoothHess + SmoothGrad (SH + SG, Ours), SmoothGrad (SG), SoftPlus Gradient (SP G), SoftPlus Hessian + SoftPlus Gradient (SP (H + G)) and vanilla Gradient (G). First order attack vectors are constructed by scaling the normalized gradient by $\epsilon$ and subtracting from the input. Second order attack vectors are found by minimizing the corresponding second-order Taylor expansions.

memorized the data and (ii) SmoothHess captures the network behaviour while the SoftPlus Hessian does not.

## F.4 Qualitative Comparison

We present a visual comparison of the interactions found by SmoothHess with those from other methods. Namely, we consider methods that can be interpreted as the quadratic term in a second-order Taylor expansion around a smooth surrogate network: SoftPlus Hessian and Swish Hessian.

We show interactions found between super-pixels of CIFAR10 test images. To this end, we utilize the Simple Linear Iterative Clustering (SLIC) [2] algorithm to segment the image into 20-25 super-pixels. We sum interactions between each pair of features in each pair of super-pixels, before visualization.

Results are shown in Figure 6 for the predicted class logit of a ResNet18 model for three CIFAR10 test images. Here, each row corresponds to a separate image. Test images are visualized in column 1. Columns 2-4 correspond to the three methods. For each image, interactions between one chosen

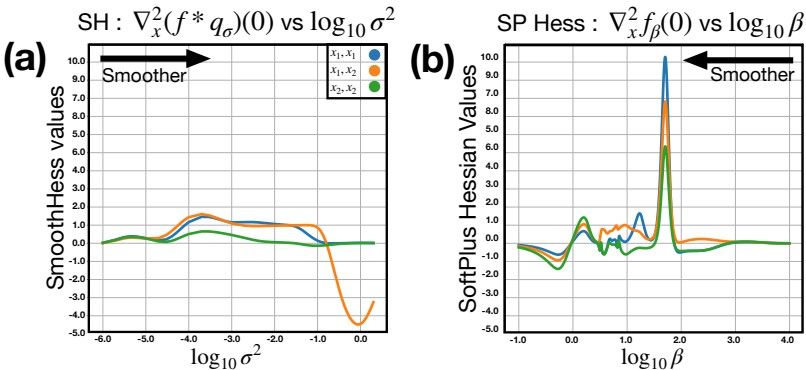

Figure 5: Three estimated Hessian elements at $x_0 = (0,0)^T$ for a 6-layer ReLU Network $f : \mathbb{R}^2 \to \mathbb{R}$ trained on the Nested Interactions dataset. **(a)** SmoothHess (SH) is estimated with isotropic covariance $\Sigma = \sigma^2 I$ using granularly sampled $\sigma^2 \in \{1e\text{-}6, \ldots, 10\}$. At minute $\log_{10} \sigma^2 < -4$ either hyper-local noisy behavior is captured, or smoothing is so negligible that the smoothed function is approximately piece-wise linear with a low-magnitude Hessian. For $\sigma^2$ ranging from $\log_{10} \sigma^2 = -4$ to $\log_{10} \sigma^2 = 0$ we see SmoothHess reflects the interactions in the dataset: Starting at $\log_{10} \sigma^2 = -4$ both $x_1 x_1$ and $x_1 x_2$ have an interaction $\approx 1$ until the interaction between $x_1 x_1$ begins to dip to 0 around $\log_{10} \sigma^2 = -1.5$. Finally around $\log_{10} \sigma^2 = -1$ the interaction $x_1 x_2$ begins to dip toward $-5$, until $\log_{10} \sigma^2 = 0$ when $\sigma^2$ is so large that samples outside the training data distribution are incorporated into SmoothHess estimation. **(b)** The Hessian of the SoftPlus smoothed function $f_\beta$ (SP Hess) is computed using granularly sampled $\beta \in \{1e\text{-}1, \ldots, 1e4\}$. Here, as $\beta$ is decreased, it is not apparent that the variety of interactions in the Nested Interactions dataset are captured, either in relative ordering or magnitude.

super-pixel (outlined in black) and each other super-pixel are visualized as a heatmap overlaid upon the image. In order to facilitate comparison across images and methods, we standardize the heatmap colorbar to range between the most negative and most positive interaction values on a per-image and method basis.

One interesting trend seen in each case is that there is a strong positive interaction between the chosen super-pixel and one other super-pixel which (a) is spatially nearby and (b) contains the class object of interest. For example, in the first row, the side-view mirror of the car positively interacts with the front wheel. In the second row, the tip of the frogs head can be seen to interact positively with the side of the head. In the third row, the upper and lower portions of the dogs front leg have a strong positive interaction.

Due to the subjectivity of this comparison, we include quantitative results above each image. Specifically, we indicate the $\mathcal{P}_{MSE}$ each method achieves within an $\varepsilon = 0.25$ ball around each image. Optimal smoothing parameters were chosen for each method for this task (see Table 3). It can be seen that SmoothHess achieves the lowest $\mathcal{P}_{MSE}$ in each case by a wide margin. Thus, SmoothHess may be the preferable option if one wishes for a visualization which best reflects the network's behaviour in an $\varepsilon = 0.25$ ball around the image.

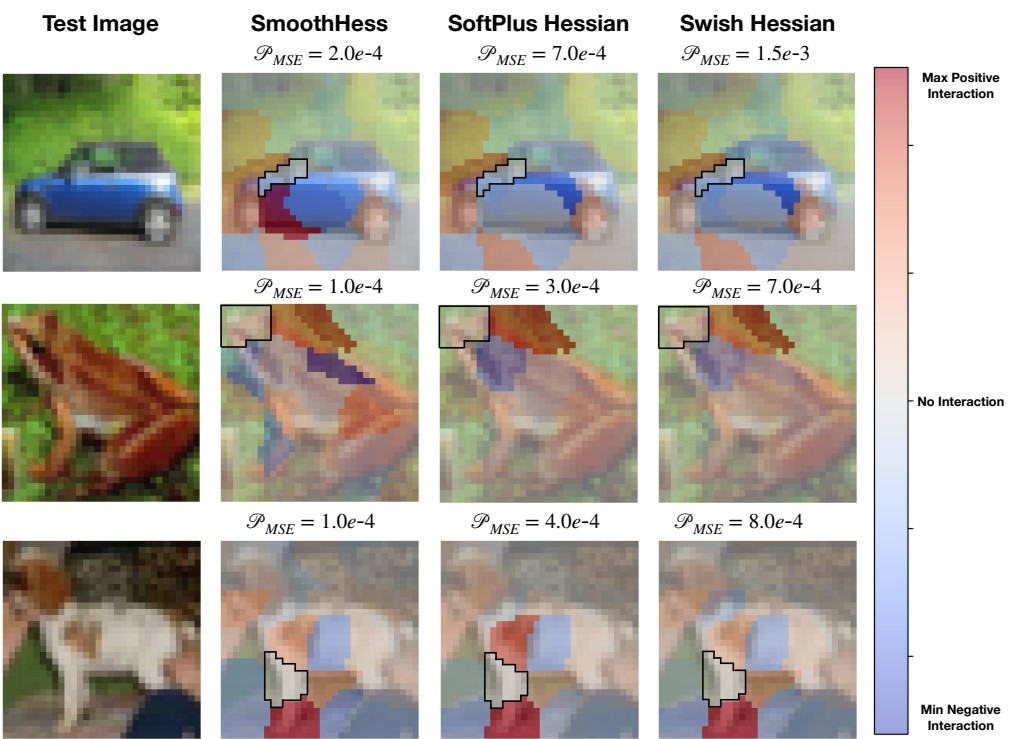

Figure 6: Visualization of interactions between super-pixels found for a ResNet18 on CIFAR10 by SmoothHess, SoftPlus Hessian and Swish Hessian. Results are shown for test images of a car, frog and dog in the first second and third rows respectively. Each image is visualised in column one. Images are segmented into 20-25 super-pixels using the SLIC algorithm [2]. Interactions are summed between each pair of features in each pair of super-pixels. We show interactions with one given super-pixel in each image, outlined in black. SmoothHess, SoftPlus Hessian and Swish Hessian interactions for this super-pixel are visualized as heatmaps overlaid upon the image in columns two, three and four, respectively. The heatmap colorbar is standardized to range between the minimum and maximum interactions on each image-method pair separately, to facilitate comparison. Quantitative $\mathcal{P}_{MSE}$ results for $\varepsilon = 0.25$ are shown above each method-image pairing, with SmoothHess achieving the lowest $\mathcal{P}_{MSE}$ in all cases.

