# OpenReview forum: "SmoothHess: ReLU Network Feature Interactions via Stein's Lemma"
_NeurIPS.cc/2023/Conference — NeurIPS 2023 poster_

### Official Review · Reviewer_bZYQ · 2023-07-05

**Soundness:** 3 good
**Presentation:** 3 good
**Contribution:** 3 good
**Rating:** 7
**Confidence:** 3

**Summary:**

For smooth functions the joint interaction between pairs of features and their impact on the function output can be modeled and quantified via second order partial derivatives. The Hessian of a neural network can therefore provide insight as to how interactions between pairs of features impact predictions made by the network. ReLU networks are piecewise linear and therefore have a zero Hessian almost everywhere, as a result a different approach to quantifying interactions is needed. One way to overcome this issue is to consider a smoothed approximation of the network. Prior works substitute ReLU for Softplus and compute the Hessian of the corresponding smooth network, however, the sensitivity to the $\beta$ parameter of the Softplus activation is pronounced. Furthermore this method affords poor control over the level of smoothing. This work considers instead estimating interactions by computing the Hessian of the convolution of the network with a Gaussian - this Hessian is referred to as SmoothHess. Furthermore, using Stein's Lemma it is shown that one can efficiently estimate SmoothHess from first order information alone using Monte Carlo simulation. Non-asymptotic bounds on the error of this estimate are also provided in Theorem 3. Experimental results demonstrating improvements of this method over existing techniques, at least in the context of relatively simple problems such as MNIST and CIFAR10, are provided.

**Strengths:**

This paper appears to provide a new and natural method with which to estimate the impact of interactions between pairs of features on the network output in the setting in which the network of the Hessian is zero almost everywhere. The method seems principled and outperforms existing methods on a number of benchmarks. The technical results appear correct although I confess I did not check them thoroughly. The paper is generally well written and clear. I am not an expert in this space so cannot say much to as to its broader context with respect to prior works.

**Weaknesses:**

One minor concern might be around how one actually chooses the covariance matrix in practice. However, choosing hyperparameters is a problem for the other methods as well it seems and furthermore this method appears to give you more direct control of the smoothing.

**Questions:**

1. Are these techniques limited to ReLU networks or could one in fact apply them to a more general class of functions for which the first and second order partial derivatives are not defined everywhere (non-smooth) and or zero almost everywhere?

2. How does one choose the covariance matrix and hence the degree of smoothing?

3. Can you identify any (potentially pathological) examples in which SmoothHess really fails to model interaction effects? In short could you provide further comment on potential failure modes of the method and highlight any warnings one should be aware of when using it?


**Limitations:**

Limitations of the work, in particular the computational complexity with regard to the input dimension, are highlighted and discussed in the conclusion. A link to code is included in the supplementary material. Potential social impacts are also discussed in the supplementary material: the authors are careful to point out that as with any method used to explain decisions made by a model one needs to treat such outputs with care and due oversight, and that it is best to work in tandem with domain experts.

---

> ### Author Rebuttal · Authors · 2023-08-10
>
> Thank you for taking the time to review our work and for the positive assessment. Below, we respond to the weaknesses and questions brought up.
>
> ---
>
> ## Weaknesses
>
> **W1: Choosing the covariance**
>
> As you mention, SmoothHess gives the user more direct control over smoothing compared to the competing SoftPlus method.
>
> In our experiment, we use isotropic covariance matrices of the form $\sigma^2 I \in \mathbb{R}^{d \times d}$. In this work we recommend a straight-forward way of choosing the level of smoothing $\sigma$. As $d \rightarrow \infty$ it can be seen that the $d$-dimensional Gaussian distribution $\mathcal{N}(0, \sigma^2 I_{d \times d})$ converges to a uniform distribution over the sphere of radius $\sigma \sqrt{d}$ [1]. in practice we have found that for the finite values of $d$ used in this paper (300,784,3072) the samples from the $d$-dimensional Gaussian $\mathcal{N}(0, \sigma^2 I)$ are indeed close to the sphere of radius $\sigma \sqrt{d}$. Thus, given one wishes to model the interactions of some network $F : \mathbb{R}^d \rightarrow \mathbb{R}^c$ in a radius $\varepsilon$-ball around a point $x_0 \in \mathbb{R}^d$ with SmoothHess, they may select $\sigma = \frac{\varepsilon}{\sqrt{d}}$. On the other hand, there is no clear cut way to choose the smoothing parameter $\beta$ for SoftPlus to draw information from a specific locality.
>
> We validate the utility of this perspective in our P-MSE experiments, shown in Table 1. In this experiment we aim to model network outputs over three neighborhoods of size $\varepsilon = 0.25, 0.50, 1.00$ for MNIST, FMNIST and CIFAR10. SmoothHess, for which $\sigma$ was chosen from a set of three values based on validation performance curated using the intuition above (described in lines 271-275), outperformed SoftPlus Hessian in each experiment. This is despite the fact that the best values of smoothing parameter $\beta$ were chosen for SoftPlus Hessian from over one-hundred options, each checked on a validation set.
>
> We additionally give an example of a use case of an alternative, non-isotropic, covariance and illustrate how such a covariance may be set. Here the goal is to focus smoothing to particular directions of input space.
>
> Consider one has a network $F: \mathbb{R}^d \rightarrow \mathbb{R}^c$ and two points $x, y \in \mathbb{R}^d$, which they wish to quantify the interactions of $F$ between. These points may have some interesting relationship: for instance $y$ may be an adversarial example for $x$ under $F$, and one may wish to find the interactions which influence this to be the case. Let us consider the sub-network $f : \mathbb{R}^d \rightarrow \mathbb{R}$, which outputs the logit (or if so desired, SoftMax probability) of the predicted class for $x$: $\text{argmax}_{c} F(x)$. Concretely, we aim to find interactions which affect $f$ between $x$ and $y$. This can be accomplished in the following way:
>
> Intuitively, we wish to estimate SmoothHess for the point between $x$ and $y$, $z = \frac{x + y}{2}$, and to focus smoothing along the direction $v = \frac{y - x}{\lVert y - x \rVert_2}$. One may use a procedure such as Gram-Schmidt to obtain an orthonormal basis for $\mathbb{R}^d$ for which the first vector is $v$ : $v, a_1, \ldots, a_{d-1} \in \mathbb{R}^d$. Let us choose a relatively large eigenvalue of $\sigma_l > 0$ for which to smooth along $v$ and small $\sigma_s \approx 0$ to smooth along the other directions. One may construct eigenvector matrix $Q = [v | a_1 | \ldots | a_{d-1}] \in \mathbb{R}^{d \times d}$ and eigenvalue matrix $\Lambda = \text{diag}(\sigma_l, \sigma_s, \ldots, \sigma_s) \in \mathbb{R}^{d \times d}$ and the covariance may be set simply by matrix multiplication $\Sigma = Q\Lambda Q^T$. Finally, SmoothHess may be fit over mid-point $z$ using $\Sigma$. One may see that this affords a higher weight to samples which are on, or close to, the line containing $x$ and $y$.
>
> Conceptually this is an example of the more general approach we allude to in lines 163-165; directions of interest can be encoded with eigenvectors and their relative weighting with eigenvalues. We will include this explanation in the camera-ready Appendix.
>
> ---
>
> ## Questions
>
> **Q1: Limitation to ReLU networks**
>
> This is a great point. As we explain in global response (G1), these techniques can be applied to any function which satisfies the assumptions of Proposition 1, namely that the function is Lipschitz continuous. Thus, as long as one has access to a first-order gradient oracle, SmoothHess may be estimated for any Lipschitz continuous function.
>
> **Q2: Choosing the covariance**
>
> We kindly direct the Reviewer to our response to W1 above.
>
> **Q3: Potential pathological examples**
>
> As illustrated in our discussion regarding the choice of covariance, SmoothHess allows for fine-grained control over the locality of the information that is taken into account when computing interactions. Specifically, this locality is determined by the eigenvectors and eigenvalues chosen for the covariance matrix. However, in our current work, we assume that the SmoothHess smoothing is determined by a Gaussian distribution. Given one wishes to represent interactions from a specific region, they should take into account how well a Gaussian smoothing can agglomerate information from that region. Future work might investigate parameterizing the SmoothHess smoothing with different distributions, which would allow for estimating interactions over a broader range of regions for a given sample.
>
> **References**
>
> [1] Roman Vershynin. High-dimensional probability: An introduction with applications in data science, volume 47. Cambridge university press, 2018.

---

> > ### Comment · Reviewer_bZYQ · 2023-08-14
> >
> > Thanks for answering my questions, I will keep my current score.

---

### Official Review · Reviewer_yoRr · 2023-07-07

**Soundness:** 3 good
**Presentation:** 4 excellent
**Contribution:** 4 excellent
**Rating:** 8
**Confidence:** 4

**Summary:**

The goal of this paper is to model the point-wise interactions between input features in a ReLU network. ReLU networks are piece-wise linear, which inhibits using Hessian to model feature interaction (since it’s zero almost everywhere). This paper proposes SmoothHess, a new method to compute feature interaction by the Hessian of a smooth surrogate of the ReLU network. The authors prove that SmoothHess can be estimated using only gradient calls, and prove non-asymptotic sample complexity bounds.

One result that’s particularly elegant is the connection between SmoothHess and SmoothGrad: combined, they define a local, second-order Taylor expansion of the network. The author designed several evaluations of the proposed method based on this connection.

The main baseline that SmoothHess is compared against is SoftPlus Hessian which is based on an alternative smooth surrogate of ReLU. These methods are evaluated as local second-order estimates of the ReLU model: they are evaluated by the accuracy of their local approximation, as well as the capability to guide adversarial attacks. SmoothHess performs best in both tasks on three standard image classification datasets. The authors also performed qualitative analysis on a regression problem from the medical domain.

**Strengths:**

Although feature interaction is a well-studied topic, I find the proposed method to be quite novel. The motivation of the problem, i.e., the piece-wise linearity of ReLU networks is very clear and the survey of related work seems thorough.
The proof sketches provided in the main paper are explained clearly, with detailed references to pieces of proofs from other work.
Overall, I find the paper to be very well-written and easy to follow. It strikes a great balance between having detailed theoretical results and clearly conveying the intuition.

**Weaknesses:**

The main weakness of the paper is the comprehensiveness of experiments. Due to computational cost of computing SmoothHess, all experiments are conducted on relatively small-scale datasets and small-ish models. Although I do not see reasons why this method would not work on larger models, it would make the paper much stronger to have empirical evidence.

I also find the qualitative analysis a bit dry. Part of this is due to my lack of familiarity with the Spirometry regression task. But I cannot confidently evaluate the significance of that result, in particular regarding what it means for interpretability of ReLU networks in general. Can this method improve tasks that can benefit from interpretability, like error detection, trust calibration, debugging, auditing, or knowledge discovery? I'm not sure.

This is a bit disappointing since interpretability is a primary motivation for this work. It would help to at least have some qualitative comparison (and with more competing methods) on more familiar datasets to at least help us understand the behavioral differences between SmoothHess and existing methods.

**Questions:**

Regarding Equation 10: if my understanding is correct, what Equation 10 does is it defines how SmoothHess (and competing methods) can be used in conjunction with gradient methods to define a local, second-order model of the underlying ReLU model. This equation is then used as the basis of experiments which evaluates the goodness of fit from several aspects. Is this understanding correct? Are there alternatives to Equation 10? Is there an alternative way to do similar experiments without Equation 10? Is it possible for this specific definition (for example by using f instead of g as the first term) to bias the results?

**Limitations:**

The authors have adequately address the limitations.

---

> ### Author Rebuttal · Authors · 2023-08-10
>
> We appreciate your enthusiasm about, and careful reading of, our work. We are especially glad to note your appreciation of the connection between SmoothHess and SmoothGrad. Below, we respond to the weaknesses and questions you have brought up.
>
> ---
>
> ## Weaknesses
>
> **W1: Generalization to larger models**
>
>
> Following your recommendation, we ran an experiment validating that SmoothHess can capture interactions for a ResNet101 trained on CIFAR10. We kindly point you toward the global response (G4) for an explanation and Table 2 of the global response pdf for results.  Thank you for bringing this concern up, this new evidence improves the generalizability of our work.
>
> **W2: Spirometry regression and qualitative comparison**
>
> We provide an updated explanation of the spirometry case-study in the global response (G2).
>
> We agree that there are many possible real-world applications for interpretability methods. As our quantitative results in Section 5.2 show, SmoothHess outperforms existing methods of estimating Hessians as well as first-order approaches, suggesting that using SmoothHess should improve downstream applications that use Hessian-based interactions. In addition, SmoothHess is very flexible in that $\sigma$ may be set to capture local interactions very effectively, at many different localities around a point. Thus, there is some reason to believe SmoothHess may outperform other methods in applications where the Hessian is important; e.g. validating that the model is picking up on interactions which domain experts know to be in the data.
>
> We agree that having additional qualitative comparisons will be beneficial. We plan to add a qualitative comparison, on image examples, with other methods to the camera-ready Appendix.  Specifically, we plan to segment images, agglomerate SmoothHess interactions within each super-pixel, and carefully compare the difference in behavior with other methods.
>
> ---
>
> ## Questions
>
> **Q1: Equation 10. second-order Taylor expansion**
>
> We appreciate this thoughtful question regarding our experimental methodology.
>
> Your understanding of Equation 10 and its use in our experiments is correct. As you hint at, the most straightforward modification to Equation 10 would be using $g(x_0)$ instead of $f(x_0)$ as the first term. Our motivation in using $f(x_0)$ for the first term stems from the fact that our goal is to evaluate the ability of the smooth surrogates _corresponding gradient-hessian explainer pair_ to model the network locally. We do not consider the quantity $g(x_0)$ (the smoothed function output) to be an explainer of interest; it does not indicate feature interactions such as SmoothHess or feature importance such as SmoothGrad. For this reason we elected to use $f(x_0)$ as the first term for all models. From our perspective this, in some sense, actually removes the bias that could be introduced by the term $g(x_0)$ which we do not aim to evaluate. This allows us to isolate the impact of the gradient-hessian pair in modeling local changes in the function.
>
> This is opposed to the different, but related, goal of _assessing the smooth surrogates second-order Taylor expansion_  in its ability to model the network, which would require replacing $f(x_0)$ with $g(x_0)$.
>
> While the experiments based on Equation 10 seem the most natural to us, other related experiments can also be run without Equation 10. For instance, as Reviewer FahQ mentioned, one may leverage the interpretation of the Hessian as a first order approximation of the gradient (or some scalarization of the gradient) to create experiments which model the gradient on the first-order, as opposed to modeling the network itself on the second-order as we have focused on. We find Equation 10 to be more in-line with our interpretability goals as generally we hope to quantify the effects of interactions for the function $f$.

---

### Official Review · Reviewer_HbQW · 2023-07-09

**Soundness:** 3 good
**Presentation:** 4 excellent
**Contribution:** 3 good
**Rating:** 6
**Confidence:** 3

**Summary:**

This paper introduces SmoothHess, a method to define a smooth approximate Hessian for networks where many components of the Hessian may be zero (e.g. ReLU networks). This can aid in tasks where interactions between features are needed, but where those components of the exact Hessian are zero. SmoothHess is defined as the convolution of the output of the network with the Gaussian. It is easy to show, using Stein's lemma, that this is equivalent to the expectation of the gradient of a perturbed output over that Gaussian distribution. They then do a number of experiments. In the first two sets of experiments (perturbation MSE and adversatrial attakcs) on three datasets to show the advantages of using SmoothHess over alternative smoothings of ReLU, such as softPlus. In both experiments, SmoothHess significantly outperforms SofPlus. They also test this on a 4 quadrants dataset with similar results. They also apply this to a real-world medical dataset of spirometry to measure feature interactions.



**Strengths:**

The paper shows the benefit of using a smoothened Hessian of ReLU networks for tasks involving feature interactions. The benefit of the method is that it is post-hoc and can be applied to the output of a trained network without a modification to the architecture. The theoretical derivations are easy to follow and well organized. The experiments also make a good case for using their construction, showing superiority to SoftPlus.
Overall, the simplicity of the method is its main strength.

**Weaknesses:**

My main reservation is how impactful the work can be. It is a neat idea, but with a rather limited scope. For example, given the expense of computing SmoothHess and the fact that we only need it for diagonal blocks of the Hessian of ReLU networks (explained next) makes the impact rather limited for me. Maybe the authors can explain or emphasize more the breadth of the problems where this would be useful or required?
One minor issue to raise is that, as I understand it, the Hessian of ReLU networks is *not* zero almost everywhere. In multi-layer networks, only the Hessian components involving weights in the same layer are zero. All cross layer Hessian terms should be nonzero. Let $W^l_i$ denote a component indexed by $i$ in the weights of layer $l$. The Hessian term
$${\partial^2 F(x) \over \partial W^l_i \partial W^k_j} $$
is in general not zero, even in ReLU networks.
In other words, the paper's claims about vanishing Hessian only hold for block diagonal components of the Hessian. They need to argue for why these components are more important than off-diagonal blocks.
Additionally, the baselines in the experiment are rather limited. SoftPlus is not the only smooth alternative to ReLU. The swish activation and similar functions could have been compared as well.


**Questions:**

1. Why only consider diagonal Hessian components, while off-diagonal blocks of ReLU networks can be non-zero?
2. How do baselines other than SoftPlus (e.g. swish) perform?
3. In the spirometry experiment, what is the conclusion? Can you trust or validate that the interactions revealed by SmoothHess are real? Does it help predict which datapoints to discard, or is it about which ones not to drop?
4. regarding the quadratic computation of SmoothHess, is there an alternative for feature interactions which is not quadratic?


**Limitations:**

The paper does discuss limitations, notably the quadratic cost in feature dimensions, which is prohibitively expensive for large models/datasets. I don't see an immediate societal impact issue.

---

> ### Author Rebuttal · Authors · 2023-08-10
>
> Thank you for taking the time to review our work. Below we respond to the weaknesses and questions brought up.
>
> ---
>
> ## Weaknesses
>
> **W1: Impactfulness and non-zero Hessian**
>
>
> We kindly wish to clarify any misunderstandings about the Hessian we are computing and its use in our work. We are interested in the Hessian used in interpretability [1,2,3,4] which is the Hessian with respect to the input $x \in \mathbb{R}^d$, with elements of the form $\frac{\partial^2 F(x)}{\partial x_i \partial x_j}, \ i,j \in \{1, \ldots, d\}$. The interpretation of this quantity is that, following the intuitions from calculus, $\frac{\partial^2 F(x)}{\partial x_i \partial x_j}$ represents the interaction between features $x_i$ and $x_j$ in affecting the models prediction for any specific point $x$, i.e. $F(x)$.
>
>
> In the particular comment the reviewer refers to the Hessian with respect to the weights ($\frac{\partial^2 F(x)}{\partial W_i^l \partial W_j^k}$), which is also a very interesting quantity  and important for optimization, e.g. second-order methods. However in this work we are not interested in this particular Hessian. In this respect, we neither need to assume nor have a block structure.
>
>
> For the fact that the Hessian (with respect to inputs) of ReLU networks is 0 almost everywhere, please see, e.g. [1]. We convey a proof sketch here: First observe that any affine function $f : \mathbb{R}^d \rightarrow \mathbb{R}^c$ of the form $f(x)  = Wx + b, \ x \in \mathbb{R}^d, W \in \mathbb{R}^{c \times d}, b \in \mathbb{R}^{c}$ has a Hessian that is 0. From here one may see that any piecewise linear function has a 0 Hessian, wherever it is defined. Further, note that the composition of piecewise linear functions is piecewise linear. Deep ReLU networks are proven to be compositions of piecewise linear functions [5].
>
> **W2 : Limited scope of baselines**
>
> Following the reviewer’s suggestion we ran new perturbation mean-squared-error (P-MSE) experiments comparing with Swish smoothed networks. We kindly point you toward the global response (G3) for an explanation of our experiment and Table 1 of the global response pdf for the results. SmoothHess outperforms Swish in all 9 permutations of the P-MSE experiment we have run.
>
> ---
>
> ## Questions
>
> **Q1 and Q2**
>
> Please see the responses to W1 and W2 above, respectively.
>
> **Q3: Regarding Spirometry Case-Study**
>
> We kindly point you toward the global response (G2) for clarification on the spirometry experiment, and the conclusions that may be drawn from it.
>
> **Q4: Regarding quadratic cost**
>
> Methods which aim to quantify second-order interactions between every pair of features for $d$-dimensional inputs are generally $\Omega(d^2)$ given each pair of $d$ elements are compared. We are not aware of works which circumvent this inherent issue.
>
> ---
>
> ## References
>
> [1] Joseph D Janizek, Pascal Sturmfels, and Su-In Lee. Explaining explanations: Axiomatic feature interactions for deep networks. J. Mach. Learn. Res., 22:104–1, 2021.
>
> [2] Michael Tsang, Dehua Cheng, Hanpeng Liu, Xue Feng, Eric Zhou, and Yan Liu. Feature interaction interpretability: A case for explaining ad-recommendation systems via neural interaction detection. In International Conference on Learning Representations, 2019.
>
> [3]Samuel Lerman, Charles Venuto, Henry Kautz, and Chenliang Xu. Explaining local, global, and higher-order interactions in deep learning. In Proceedings of the IEEE/CVF International Conference on Computer Vision, pages 1224–1233, 2021.
>
> [4] Sahil Singla, Eric Wallace, Shi Feng, and Soheil Feizi. Understanding impacts of high-order loss approximations and features in deep learning interpretation. In International Conference on Machine Learning, pages 5848–5856. PMLR, 2019.
>
> [5] Randall Balestriero and Richard Baraniuk. Mad max: Affine spline insights into deep learning, 2018.

---

> > ### Comment · Reviewer_HbQW · 2023-08-16
> > **Thanks fro clarifications**
> >
> > Thank you for clarifying which Hessian you are using, which I had misinterpreted. For Hessian w.r.t. the input, it is clear that off-diagonal terms are zero for ReLU networks.
> >
> > Also, thanks for the Swish experiments. I will increase my score.

---

### Official Review · Reviewer_GnHq · 2023-07-10

**Soundness:** 3 good
**Presentation:** 2 fair
**Contribution:** 2 fair
**Rating:** 6
**Confidence:** 2

**Summary:**

The paper propose to compute Hessian of a ReLU neural network by a convolution of the network function with a Gaussian distribution function. It's equivalent (or cover) some prior work, e.g., SmoothGrad. The method is generally easy to understand, without modifying the network structure. A non-asymptotic bound on the sample complexity is provided.

**Strengths:**

Strengths:
-- The proposed smooth-surrogate is easy and straightforward to understand, the computation method is somewhat novel using Stein's Lemma, following [40] with a mild extension to all Lipschitz continuous functions, which addresses the challenge posed by the zero Hessian of piecewise-linear ReLU networks. The paper provides a non-asymptotic bound on the sample complexity of the estimation procedure.

-- Proposed SmoothHess can be applied post-hoc and does not require any modifications to the ReLU network architecture. This allows analysis and interpretation of the interactions of existing ReLU neural networks without retraining or redesigning the model.

-- Efficient sampling: SmoothHess utilizes an efficient sampling algorithm, which only requires network gradient calls, making it computationally feasible to large-scale networks and complex datasets.

**Weaknesses:**

Weaknesses:
-- Scope limited to ReLU networks: While SmoothHess addresses the challenge of zero Hessian in ReLU networks, its applicability is limited to this specific activation function. It would be useful to discuss the potential generalizability of the method to other types of neural networks with different activation functions.

-- Generalize beyond simple datasets: SmoothHess can be applied to large models and datasets with its simple computations, it would be beneficial to clarify the generalizability of SmoothHess and the potential limitations in terms of dataset diversity.

**Questions:**

Some notations are confusing, e.g., caption of figure 2, is it $\log(10\sigma)$ or $\log_{10}{\sigma}$?

also the text, "Aside from at minute", what does "minute" mean here?

**Limitations:**

See above

---

> ### Author Rebuttal · Authors · 2023-08-10
>
> Thank you for taking the time to review our work. Below we respond to the weaknesses and questions brought up.
>
> ---
>
> ## Weaknesses
>
> **W1: Limitation of scope to ReLU**
>
> We appreciate the comment as we feel this is an important point to clarify in our work. Our method is not limited only to ReLU networks and in fact it is of use in far broader settings. We focus our presentation on ReLU networks because, for their logits and internal neurons, smoothing _is required_ for the computation of non-zero Hessians. Proposition 1 states an equivalency between the expectation our method estimates and SmoothHess for _any Lipschitz continuous function_. Thus our method is applicable to any Lipschitz continuous network or model for which one has access to the gradients (i.e. can backpropagate through). This covers a wide range of networks, for instance networks which may be composed of convolutional layers, fully connected layers, max pooling, common activation functions such as sigmoid, skip connections and batch normalization among other frequently used network building blocks [1].
>
> SmoothHess still provides useful information for Lipschitz networks which have a non-zero Vanilla (unsmoothed) Hessian. Although the Vanilla Hessian does provide information pertaining to the behavior of such functions, the locality of this information is infinitesimally small. On the other hand, SmoothHess has a smoothing parameter $\Sigma$ which may be chosen to model the interactions over neighborhoods of varying size, providing the user great flexibility.
>
> We plan to emphasize this point in our camera-ready paper.
>
> **W2:  Generalization to larger datasets and models**
>
> Indeed, the input size matters, which we point out in the limitations section. Specifically, we note that the most computationally expensive aspect of SmoothHess is the outer-products, which are $\mathcal{O}(d^2)$. For large $d$, this becomes expensive. We gave the example of ImageNet, which has $d^2 \approx 10^{10}$. However, we do not view this as a limitation relative to other works, since interaction methods are generally $\Omega(d^2)$ [2,3,4], as each pair of $d$ features must be compared. In our work, we have run SmoothHess for inputs of up to size $d=3072$, although we have not exhaustively checked the limit for which SmoothHess is feasible. From a qualitative perspective of dataset diversity, we have managed to capture feature interactions for a number of different datasets: MNIST, FMNIST, CIFAR10, real world Spirometry regression, synthetic Four Quadrant and the synthetic Nested Interactions dataset found in Appendix F.3.
>
> We also wish to highlight the new experimental results reported in the global response (G4), which provide evidence that SmoothHess can generalize to larger models. Specifically, our results show that SmoothHess can capture feature interactions for a ResNet101. Table 2 of the global response pdf shows our results for this experiment.
>
> ---
>
> ## Questions
>
> **Q1: Figure 2 caption.**
>
> Thank you for pointing this out. We mean $log_{10} \sigma$ and will add the subscript to make this more explicit in the camera ready version.
>
> **Q2: Meaning of “minute”**
>
> Here minute means “very small”.
>
> ---
>
> ## References
>
> [1] Gouk, Henry, et al. "Regularisation of neural networks by enforcing lipschitz continuity." Machine Learning 110 (2021): 393-416.
>
> [2] Joseph D Janizek, Pascal Sturmfels, and Su-In Lee. Explaining explanations: Axiomatic feature interactions for deep networks. fJ. Mach. Learn. Res., 22:104–1, 2021.
>
> [3] Michael Tsang, Sirisha Rambhatla, and Yan Liu. How does this interaction affect me? inter- pretable attribution for feature interactions. Advances in neural information processing systems, 33:6147–6159, 2020.
>
> [4] Dhamdhere, Kedar, Ashish Agarwal, and Mukund Sundararajan. "The shapley taylor interaction index." arXiv preprint arXiv:1902.05622 (2019).

---

> > ### Comment · Reviewer_GnHq · 2023-08-19
> >
> > I have read the authors' reply. Many thanks to the authors for the clarifications. My concerns and questions are mostly satisfactorily addressed. I have no further questions and will remain positive to this paper.

---

### Official Review · Reviewer_FahQ · 2023-07-29

**Soundness:** 3 good
**Presentation:** 2 fair
**Contribution:** 3 good
**Rating:** 6
**Confidence:** 2

**Summary:**

The work proposes an approach for measuring feature interactions for neural networks that have no
useful second order gradients or Hessians (notably those employing ReLU). The approach convolves a
smooth normal with the network as a proxy with usable Hessians and then offers a means of
estimating them using sampling of first-order gradients, thereby making it more tractable than
methods that rely on computing Hessians naively. Experiments evaluate the proposed method and
several adaptations of existing methods for computing 1) the 3rd-term (second order gradient) of
the Taylor expansion of the output of a neural network around an input, and 2) for adversarial
inputs via optimization using (1). In these experimental terms, the proposed approach (aided by
SmoothGrad for the 2nd Taylor term) performs better than those based on only the first order
gradient or those that use include both first and second order gradient but proxy ReLU models using
SmoothPlus activation.

**Strengths:**

+ Almost strictly better experimental results than the compared-to methods.

+ Hessians for networks which don't naively have them due to technicalities should be generally
  useful in a wide variety of applications.

**Weaknesses:**

- Experimentation focuses on agreement between a network and one evaluated as per Taylor expansion
  using SmoothHess. While this may be a fairly general test, it does not offer any insight into
  benefits in cases where the method would actually be useful to use, with no trivial alternatives.
  Presumably in most cases we can already evaluate the network regardless of Hessian non-existence
  so at surface level there is no need for Taylor expansion and thus the Hessian of the smooth
  proxy. The adversarial attack is also based on use of Taylor expansion of smoothed model as proxy
  so it does not improve this problem. The final case-study does present interaction results but is
  difficult to follow (see questions below). I suggest, then, that the paper include evaluation of
  the methods that focus on the Hessian, instead of network output. One example is to test
  adversarial attacks on gradient-based attributions by gradient descent on the gradient (i.e.
  using Hessian). Compare to other methods for attacking attributions.

- Tradeoffs with respect to the parameters are not explored or described. The use of smoothing
  serves to increase the impact of network regions further away from each point but smoothing is a
  double-edged sword. If smoothing (and subsequent sampling) is insufficient compared to the sizes
  of linear network regions, the results may not be useful. On the other hand, large enough
  smoothing reduces the network to a constant.

  One impact of this tradeoff is how fast gradient descent can arrive at good solutions. Thus in
  addition to the above suggestion of testing usefulness of the Hessian, I also suggest that the
  effect of the parameters (or smoothness) have on that usefulness. For example, sufficiently
  significant smoothness I expect to result in useless gradients of gradients for attacking
  attributions.

Smaller things:

- What is δ (without subscript) in Equation 8a? Did you mean δᵢ ?

- On line 117, the expectation and perturbation are presumed not significant to the point that the
  Hessian is 0 almost everywhere.

- By "second-order interactions" did you mean interactions via second-order gradients? Interactions
  seem already second-order so saying second-order interactions is suggesting third-order
  gradients.

- In Table 1, are the two best (bolded) results in the 4th numerical column identical or is it just
  that the printout identical?

- Around line 255, you write "We denote Δ* ... as the output after x₀ is attacked". This may be a
  bit confusing as I think you mean Δ* as the perturbed input, not the network output. Is that
  right?

- Line 349 writes "Interestingly, the smaller kernel width constraints the interactions to features
  within a small locality." Why is that interesting? Isn't that expected?

- Typo near "between between".

- The Adversarial Attacks paragraph on line 254 and Equation 12a is missing the target class of the
  attack.


**Questions:**

- Conceptually, when sampling as per MC-estimation, if samples end up not spanning more than 1
  linear region of the original network, would the estimate be zero in the same sense as second
  order gradient is zero?

- What is the impact of smoothness/covariance on sample bound as per Equation 9?

- Does "SP H + G" mean "SoftPlus (Hessian + Gradient)" or "(SoftPlus Hessian) + (ReLU Gradient)"?
  Line 285 suggests the latter but the label/caption suggest the first. If the first, the
  comparison may not be as fair as it could be.

- Example of Figure 3 is difficult to understand. Why are the arrows labeled "interaction" crossing
  from one method to another? Also, SmoothGrad vs. SmoothHess is indicated by both color and the
  separate graphs? Did you mean that green/red are indicators of positive or negative interactions
  for both of those methods or for one respectively? Or perhaps the second of each pair of graphs
  shows the interaction of the other timesteps with the first timestemp only? Also, what is the
  kernel width for the example shown in Figure 3?

- The covariance matrix is set in a specific way in the text but the benefit of the option to set
  it otherwise are mentioned. How would one go about setting it alternatively and for what
  purposes?

**Limitations:**

The work describes experiments with adversarial attacks which could use some mention with relation
to negative societal impact.

---

> ### Author Rebuttal · Authors · 2023-08-10
>
> Thank you for reviewing our work. **We've addressed your main concerns in our official rebuttal. Other concerns have been addressed in a comment due to the character limit.**
>
> ---
>
> **W1  Experimentation Focus**
>
> We emphasize that, while we agree as to the potential for use in optimizations, SmoothHess is foremost meant to quantify feature interactions. Assessing how well any given interpretability method explains network behavior is non-trivial: it is often subjective and dependent on expert knowledge. A number of techniques have been introduced to evaluate univariate methods[1,2]. Note that these methods don’t necessarily correspond to real world uses. They are used as a quantitative proxy for interpretation quality, but are not the final goal.
>
> As SmoothHess and SmoothGrad together define a second-order Taylor expansion we developed P-MSE and the adversarial attack experiment as intuitive quantitative measures of interaction quality, using this Taylor expansion as a proxy. Low values of P-MSE may indicate that the average interactions are captured. Successful adversarial attacks may indicate that extremal interactions are captured.
>
> We find your experiment valuable, but believe it may be subject to the same comments given regarding our adversarial attack experiment. While the use of SmoothHess to attack the function f presupposes access to f, the use of SmoothHess to attack $\nabla f$ presupposes access to $\nabla f$, which is used for SmoothHess computation. We believe both experiments are interesting, but note that the adversarial attack on f may be more in line with our goal of interpreting interactions for f. This is as opposed to interpreting first order importance for $\nabla f$, which is also interesting. We leave this for future work.
>
> As stated above, the main application of SmoothHess is to be used to interpret network behavior. This is exemplified by the spirometry case study, which we clarify in the global response(see G2).
>
> **W2  Smoothing Tradeoffs**
>
> We agree with this need and note that we consider the benefits of SmoothHess to be highlighted from this perspective.
>
> First, from our viewpoint, the trade-off from smoothing is not inherent, i.e. “smoothing level $\sigma$ is too high for a given point/network”, but rather predicated on the information one is searching for. E.g., assume one is given a point $x \in \mathbb{R}^d$ and network $f : \mathbb{R}^d \rightarrow \mathbb{R}$, and they wish to find the interactions in the radius 0.001 ball: $B_{0.001}(x) \subset \mathbb{R}^d$. If there exists a linear region $Q \subseteq \mathbb{R}^d$ s.t. $B_{0.001}(x) \subseteq Q$ then, from the Hessian perspective of interactions, there is _no interaction occurring_ in $B_{0.001}(x)$. In this scenario using a tiny $\sigma$ would actually be appropriate for answering the users’ question, _precisely because it would result in a near zero Hessian._
>
> Herein lies the relative benefit of our method over SoftPlus. Given a ball of radius r over which a user wishes to model interactions, $\sigma$ can be chosen so that SmoothHess draws samples from near the boundary of this ball. This is because the d-dimensional Gaussian with covariance $\sigma^2 I$ converges to a Uniform distribution over the sphere of radius $\sigma \sqrt{d}$ as d goes to $\infty$ [3]. Thus the covariance may be set to $\frac{r}{\sqrt{d}}$, ensuring that SmoothHess uses samples near this sphere. This is explained in lines 171-177.
>
> Although this convergence occurs as d goes to $\infty$ we have observed that it is close to reality with the datasets used, d = 300,784,3072. We validate this approach in our P-MSE experiment: SmoothHess with $\sigma$ chosen based upon $\frac{r}{\sqrt{d}}$ achieves the best P-MSE over competing methods. Alternatively, the smoothing parameter $\beta$ for SoftPlus has no clear-cut connection to a locality, for exploitation.
>
> Lastly, we point to the Nested Interactions Experiment in App. F.3, in which we assess the ability of SmoothHess to pick up interactions as $\sigma$ is varied. We sample a set of points $x_1, \ldots, x_N \in \mathbb{R}^2 $ uniformly from $[-2,2] \times [-2,2]$ and set targets $y(x)$ by: $x \in B_{0.6}(0) \implies y(x) = \frac{1}{2} x_1^2 + x_1 x_2$, $x \in B_{1.2}(0) \backslash B_{0.6}(0) \implies y(x) = x_1x_2$  $x \in \mathbb{R}^2 \backslash B_{1.2}(0) \implies y(x) = -5x_1x_2$. We train a network to “memorize” this dataset.
>
> Thus, we have created “ground truth” interactions for _network behavior_ which change with distance to the origin. We estimate SmoothHess at $x_0 = 0$ with $\sigma \in \{1e-6, \ldots, 10 \}$ and plot the interactions in Figure 5a. The interaction between $x_1$ and $x_2$ should be $\approx 0$ within a single region. At slightly larger localities one should expect to capture the noisy behavior of the network. Then an interaction of $1$ should occur and then a decrease to $-5$. Our results in Figure 5 capture this behavior. The behavior at largest $\sigma$ is due to the many samples incorporated outside of the train distribution.
>
> ---
>
> **S4 Precision**
>
> This is due to precision. Checking with higher precision, we found SmoothHess P-MSE = 4.9e-8 and SoftPlus Hess P-MSE = 5.5e-8. Thus, SmoothHess achieves lowest P-MSE in all cases. We will add this to the caption.
>
> **Q3 Fair comparison**
>
> We use SP H + SP G. We feel this choice is more fair. Table 1 in the response pdf shows SP G significantly outperforms (ReLU) G. We will make this clear in the final version.
>
> ---
>
> **References**
>
> [1] Hooker, Sara, et al. "A benchmark for interpretability methods in deep neural networks." Advances in neural information processing systems 32 (2019).
>
> [2] Chen, Jianbo, et al. "Learning to explain: An information-theoretic perspective on model interpretation." International conference on machine learning. PMLR, 2018
>
> [3] Roman Vershynin. High-dimensional probability: An introduction with applications in data science, volume 47. Cambridge university press, 2018.

---

> > ### Author Response · Authors · 2023-08-10
> > **Re: Supplementary Clarifications**
> >
> > In the Official Rebuttal, we have referenced a comment we would be making below our official response to the Reviewer, clarifying more points, which we have already prepared. However, upon further re-reading of the guidelines and new 6,000 character limit rule, we are now uncertain if providing this would be a violation of any policy and thus decided not to post.
> >
> > Thank you for your thoughtful questions, which we have tried to answer within the given character limit. If you have any specific questions please feel free to reach out during the discussion phase and we would be happy to respond.

---

> > > ### Comment · Reviewer_FahQ · 2023-08-18
> > >
> > > Response offers reasonable additional discussion around my main two concerns but has not moved its overall position and my evaluation.

---

> > > > ### Author Response · Authors · 2023-08-18
> > > > **Re: Additional Information**
> > > >
> > > > **We kindly thank the Reviewer for their response. We hope that this comment will address any remaining concerns; if not, we are happy to provide any additional information as needed.**
> > > >
> > > > ---
> > > >
> > > > ## Smaller things
> > > >
> > > > **S1, S7 Typo**
> > > >
> > > > We will change to $\delta_i$, and remove the second between, in the final version.
> > > >
> > > > **S2 Expectation**
> > > >
> > > > The space where the Hessian is 0 has measure 1 under the Gaussian distribution, from here the expectation in line 117 follows.
> > > >
> > > > **S3 Second order**
> > > >
> > > > We mean second order interactions via second order gradients. We are using the terminology: X-order interaction conveys an interaction between X features [1].
> > > >
> > > > **S5 Adv. attack phrasing**
> > > >
> > > > Thank you for catching this. We will change to “the perturbed input” in the camera ready version.
> > > >
> > > > **S6 Kernel width**
> > > >
> > > > We agree that this behavior is to be expected. We mean that this provides some qualitative evidence that the interactions SmoothHess picks up “make sense”; i.e., it is following the intuition you have for why this result is to be expected: smaller kernels result in a smaller receptive field for neurons in each layer. We will clarify this in the camera-ready version.
> > > >
> > > > **S8 Missing target**
> > > >
> > > > $\tilde{f}$ in Eq. 12a is a real valued function: the Taylor expansion around the predicted class SoftMax score for the point $x_0$ (lines 261-262). Thus, no target is required in Eq 12a. We will make this more clear in the final version.
> > > >
> > > > ---
> > > >
> > > > ## Questions
> > > >
> > > > **Q1 One region**
> > > >
> > > > Let us assume we are estimating SmoothHess for a point $x_0 \in \mathbb{R}^d$ and function $f : \mathbb{R}^d \rightarrow \mathbb{R}$ using covariance $\Sigma \in \mathbb{R}^{d \times d}$ . We have $N$ IID samples $\delta_1, \ldots, \delta_n \sim \mathcal{N}(0, \Sigma)$. Further, each perturbed point $x_0 + \delta_i$ lies in the same linear region $Q \subseteq \mathbb{R}^d$. Our (unsymmetrized) estimator would be $\hat{H}^\circ_n(x_0,f, \Sigma) = \frac{1}{n} \sum_{i=1}^n \Sigma^{-1} \delta_i [\nabla_x f(x_0 + \delta_i)]^T$. Now, because we have assumed each $x_0 + \delta_i$ lies in the same linear region, the gradient of $f$ is the same at each point:  $\nabla_x f(x_0 + \delta_i) = G, \forall i \in \{1, \ldots, n\}$ for some constant $G \in \mathbb{R}^d$. Thus our estimator may be written as $\hat{H}^\circ_n(x_0,f, \Sigma) = (\Sigma^{-1}) (\frac{1}{n} \sum_{i=1}^n \delta_i)(G^T)$. Here the terms in the left and right parenthesis are constants. Note the term in the middle parentheses is the sample mean of an R.V. distributed from $\mathcal{N}(0,\Sigma)$. Thus, if one has many samples the middle term will be close to, but not necessarily exactly, $0$ and accordingly so will $\hat{H}^\circ_n(x_0,f, \Sigma)$.
> > > >
> > > > It is not necessarily the case that $\lim_{n \rightarrow \infty} \hat{H}^\circ_n(x_0,f, \Sigma) = 0$, as eventually one would sample points in different regions: $x_0 + \delta$ s.t. $\nabla f(x_0 + \delta) \neq G$. However, It can be seen from this formulation that SmoothHess would give the $0$ matrix, in the infinite limit of samples, for a linear network $f$.
> > > >
> > > > **Q2 Bound**
> > > >
> > > > We leave the explicit incorporation of the covariance into our bound to future work.
> > > >
> > > > **Q4 Figure 3.**
> > > >
> > > > SmoothHess can be characterized as a second-order extension of SmoothGrad, and SmoothGrad values can be calculated at the same time as SmoothHess with minimal additional computational cost. Therefore, in the Spirometry experiment we use SmoothGrad to first identify first-order feature importance values, and then use SmoothHess to identify second-order interactions. In Figure 3, the left/right barplots for each spirometry sample represent the SmoothGrad/SmoothHess (respectively) attributions for 0.5sec blocks of features.
> > > >
> > > > In our example we are interested in the interactions for the first 0.5sec feature block, which is identified as being a top feature using SmoothGrad. Therefore the SmoothHess plot shows interactions between the first 0.5sec feature block and every other feature (i.e. the first row in the $d \times d$ feature interaction matrix). The red arrow points to the top interacting feature (arrow endpoint) with respect to the first feature (arrow source).
> > > >
> > > > The red/green bar colors are meant to indicate positive/negative values of SmoothGrad and SmoothHess; we will change this to a single color to reduce confusion. The kernel width for the example in Figure 3 is 200; this and other model details are listed in Appendix E.1.
> > > >
> > > > Thanks for your feedback, we will add detail to the figure caption to make this more clear.
> > > >
> > > > **Q5 Alt. cov.**
> > > >
> > > > We kindly direct you to paragraphs 4-7 of response W1 to Reviewer bZYQ for the answer to this question.
> > > >
> > > > ## References
> > > >
> > > > [1] Tsang, Michael, Dehua Cheng, and Yan Liu. "Detecting statistical interactions from neural network weights." arXiv preprint arXiv:1705.04977 (2017).

---

### Official Review · Reviewer_3HE6 · 2023-08-01

**Soundness:** 4 excellent
**Presentation:** 3 good
**Contribution:** 3 good
**Rating:** 7
**Confidence:** 4

**Summary:**

The author introduces a novel method to approximate the Hessian of a smoothed
loss function at a specific point. By incorporating a variation of Stein’s lemma,
they cleverly estimate the gradient over a smoothed surrogate of a ReLU network
at that point. To gauge the method’s effectiveness, they measure its performance
by testing the normalized L2 distance between the loss function and its first-
order and second-order approximations within a ball centered around the given
point.

**Strengths:**

The method offers a straightforward and comprehensible approach, presenting
a powerful and novel tool for approximating the Hessian of a ReLU network at
a specific point. The author’s analysis is comprehensive and remarkably easy
to follow. The experiments conducted in the paper appear relevant, and the
obtained results show promising potential.

**Weaknesses:**

While the ideas and methods of the paper are solid, I came across some read-
ability issues.

1.
The statement of Theorem 3 (line 211) is confusing. The real positive
$\delta$ is used without declaration, while the random variables $\delta_1,\ldots,\delta_n$ are declared. This makes the reader wonder how one can compare a probability with a random vector, and what is the meaning of this inequality which is essentially a random variable itself. I had to read the proof in the appendix to fully understand the statement of the theorem. I think using a different symbol for $\delta$ might increase readability.

2.
I would add the definitions of the oracles used in the article for clarity.

**Questions:**

1. line 103 "An general L-hidden..." is a typo?

2. line 287 "with the non-smooothed..." is a typo?

**Limitations:**

The article address the time and space complexity of the method in the presence of a high-dimensional input.

---

> ### Author Rebuttal · Authors · 2023-08-10
>
>
> We appreciate your positive assessment of our work. Below, we address the weaknesses and questions that you have brought up.
>
>
> ## Weaknesses
>
> ---
>
> **W1: Regarding the statement of Theorem 3.**
>
> Thank you for your careful reading of the statement of Theorem 3. We agree with the reviewer’s suggestion. In the camera ready version, we will change the real positive $\delta$ to $\gamma$ in the statement and proof of Theorem 3. We will also explicitly declare $\gamma$ in the statement of Theorem 3.
>
> **W2: Regarding oracle definitions**
>
> A: We will add the definitions of both oracles to the camera-ready version. Namely, a zero-th order oracle is a function call. That is, it is an oracle that, given a function $f : \mathbb{R}^d \rightarrow \mathbb{R}$ and input $x \in \mathbb{R}^d$, returns $f(x)$. In the context of our work, this amounts to a neural network forward pass. Likewise, a first-order oracle is an oracle which, given $x$, returns $\nabla_x f(x)$. In the context of our work, this amounts to a network backpropagation call.
>
>
> ## Questions
>
> ---
>
> **Q1 and Q2: Typos: “An general L-hidden … “, “non-smoothed”**
>
> Thank you for catching these. We will change the sentence to “A general L-hidden … “ and “non-smoothed” to “unsmoothed” in the camera ready version.

---

### Author Rebuttal · Authors · 2023-08-10

We would like to thank the reviewers for their thoughtful questions and comments, which have helped us to improve our work.


---

## Recurring Questions

Below we respond to recurring questions among the reviewers. We also give individual responses to all the reviewer questions under each reviewer’s comments

**(G1) Generality of results:** Reviewers GnHq and bZYQ raised questions about the generality of our results for non-ReLU networks or functions. As highlighted in  Proposition 1 in the paper, SmoothHess may be estimated for any Lipschitz continuous function, given one can query the gradient. We focus our presentation on ReLU networks as smoothing _is required_ in order to evaluate a non-zero Hessian. SmoothHess is still useful for Lipschitz continuous networks which have a non-zero Hessian. The reason for this is that this Hessian reflects a very small area, while SmoothHess can incorporate information from larger regions by adjusting the smoothing parameter $\Sigma$.

**(G2) Spirometry experiment:** Reviewers yoRr, FahQ, and HbQW expressed a need for clarification of the spirometry regression case-study. The purpose of the spirometry experiment is to qualitatively evaluate whether SmoothHess can be used to understand how ReLU models make predictions. FEV$_1$, a metric frequently used to evaluate lung health, is always calculated within the first 2 seconds of the spirometry curve, and _is known to be strongly affected by the presence of coughing_ [1]. In each curve, coughing may occur during, or after, the first 2 seconds. The time when coughing occurs, indicated by plateaus in the spirometry curves in Figure 3, should be an important signal for any model trained to predict FEV$_1$.

We aim to test whether SmoothHess can detect interactions between the initial segment of the curve and segments that indicate coughing (as characterized by plateauing in the curve). To this end, we train a network to predict FEV$_1$ for spirometry samples that exhibit coughing. When we apply SmoothHess in Figure 3, we find that indeed the strongest interactions for the first 0.5 second curve segment are where plateaus occur in the curve, indicating coughing. We will add this clarification to the camera-ready revision.

---

## New Results

We also present new results in the global response pdf, which we explain below.

**(G3) Table 1**: Reviewer HbQW asked for a comparison with the Hessian and gradient of a Swish smoothed network. Swish is a smooth activation function that may be parameterized by a smoothing value $\beta$ [2]. We compare the P-MSE of the Hessian + Gradient and Gradient of the Swish smoothed network with our previous results for the predicted class logit on MNIST, FMNIST, and CIFAR10, over three different radii $\varepsilon = 0.25, 0.50, 1.00$. Following the methodology used for SoftPlus in our work, we select the best value of $\beta$ for Swish based on the validation set performance from a set of over one-hundred options. This is done separately for Swish Hessian + Swish Gradient (Swish (H + G)) and Swish Gradient (Swish G), each dataset, and for each locality.

Our new results indicate that the second-order Taylor expansion using SmoothHess and SmoothGrad (our method) outperforms both Swish (H + G) and Swish G in all cases. Swish (H + G) has a particularly poor performance on CIFAR10. We will include these results in our camera-ready Appendix.

**(G4) Table 2**: Reviewers yoRr and GnHq were both interested in the ability of  SmoothHess to generalize to larger networks. We train a ResNet101 ($\approx 44.5$M parameters) on CIFAR10 and calculate P-MSE results for SmoothHess + SmoothGrad (our method), SmoothGrad, and the vanilla (unsmoothed) gradient at three radii $\varepsilon = 0.25, 0.50, 1.00$. Our method significantly outperforms both SmoothGrad and the vanilla gradient in each case. This provides evidence that SmoothHess can model interactions in large networks such as ResNet101, improving upon first-order models.

Due to the size of ResNet101, we did not have the time to evaluate SoftPlus or Swish results for ResNet101. This is due to the expensive validation procedure (mentioned in G3) that must be used for SoftPlus and Swish. In our original P-MSE experiment, we validate over $100$ different values of SoftPlus parameter $\beta$ on a held out set, before selecting the best to use on the test set. This is because there is no clear-cut way to choose $\beta$ given a desired locality for smoothing. This is opposed to our procedure for choosing $\sigma$ for SmoothHess before computing P-MSE, which exploits well known properties of the Gaussian to curate a set of only three values of $\sigma$ for validation (as described in lines 171-177 and 269-275).

We plan to validate the best values of $\beta$ for SoftPlus and Swish on the ResNet101 and add the results to Table 2, which we will include in our camera-ready Appendix. We note that there is little reason to suspect SoftPlus or Swish will be competitive with SmoothHess. As described in (G3), SmoothHess outperforms the Swish Hessian in all cases. Further, our original results show SmoothHess outperforming SoftPlus on 17 of the 18 P-MSE experiments, with one tie. However, Reviewer FahQ inquired as to the printing precision (number of significant figures) of the reported tie, and, once checked, we found that SmoothHess achieved a lower P-MSE.

---

If any reviewers have further questions or wish us to elaborate on our responses, we would be happy to address them over this upcoming discussion period.

---

## References

[1] Luo AZ, Whitmire E, Stout JW, Martenson D, Patel S. Automatic characterization of user errors in spirometry. Annu Int Conf IEEE Eng Med Biol Soc. 2017 Jul;2017:4239-4242. doi: 10.1109/EMBC.2017.8037792. PMID: 29060833.

[2] Ramachandran, Prajit, Barret Zoph, and Quoc V. Le. "Searching for activation functions." arXiv preprint arXiv:1710.05941 (2017).

---

### Decision · Program_Chairs · 2023-09-21

**Decision:**

Accept (poster)

**Comment:**

Recent work has considered studying interpretability by modeling feature interactions through the Hessian of a neural network (output-to-feature). As the Hessian of ReLU networks, being piecewise-linear, (generically) vanishes, the authors propose to study ReLU networks by convolving with a Gaussian.

Smoothing techniques of similar nature are used throughout the optimization literature. The concrete application proposed in the work, however, has been found novel and significant by the reviewers, who have also appreciated the clarity and the simplicity of the approach. Please make sure to address the important points raised by the reviewers; in particular, applicability of the proposed method beyond relatively small-scale settings.